# ADMM without a Fixed Penalty Parameter: Faster Convergence with New Adaptive Penalization

**Yi Xu[†], Mingrui Liu[†], Qihang Lin[‡], Tianbao Yang[†]**
[†]Department of Computer Science, The University of Iowa, Iowa City, IA 52242, USA
[‡]Department of Management Sciences, The University of Iowa, Iowa City, IA 52242, USA
{yi-xu, mingrui-liu, qihang-lin, tianbao-yang}@uiowa.edu

## Abstract

Alternating direction method of multipliers (ADMM) has received tremendous interest for solving numerous problems in machine learning, statistics and signal processing. However, it is known that the performance of ADMM and many of its variants is very sensitive to the penalty parameter of a quadratic penalty applied to the equality constraints. Although several approaches have been proposed for dynamically changing this parameter during the course of optimization, they do not yield theoretical improvement in the convergence rate and are not directly applicable to stochastic ADMM. In this paper, we develop a new ADMM and its linearized variant with a new adaptive scheme to update the penalty parameter. Our methods can be applied under both deterministic and stochastic optimization settings for structured non-smooth objective function. The novelty of the proposed scheme lies at that it is adaptive to a local sharpness property of the objective function, which marks the key difference from previous adaptive scheme that adjusts the penalty parameter per-iteration based on certain conditions on iterates. On theoretical side, given the local sharpness characterized by an exponent $\theta \in (0, 1]$, we show that the proposed ADMM enjoys an improved iteration complexity of $\widetilde{O}(1/\epsilon^{1-\theta})$[1] in the deterministic setting and an iteration complexity of $\widetilde{O}(1/\epsilon^{2(1-\theta)})$ in the stochastic setting without smoothness and strong convexity assumptions. The complexity in either setting improves that of the standard ADMM which only uses a fixed penalty parameter. On the practical side, we demonstrate that the proposed algorithms converge comparably to, if not much faster than, ADMM with a fine-tuned fixed penalty parameter.

## 1 Introduction

Our problem of interest is the following convex optimization problem that commonly arises in machine learning, statistics and signal processing:

$$\min_{\mathbf{x}\in\Omega} \quad F(\mathbf{x}) \triangleq f(\mathbf{x}) + \psi(A\mathbf{x}) \tag{1}$$

where $\Omega \subseteq \mathbb{R}^d$ is a closed convex set, $f : \mathbb{R}^d \to \mathbb{R}$ and $\psi : \mathbb{R}^m \to \mathbb{R}$ are proper lower-semicontinuous convex functions, and $A \in \mathbb{R}^{m \times d}$ is a matrix. In this paper, we consider solving (1) by alternating direction method of multipliers (ADMM) in two paradigms, namely deterministic optimization and stochastic optimization. In both paradigms, ADMM has been employed widely for solving the regularized statistical learning problems like (1) due to its capability of tackling the sophisticated structured regularization term $\psi(A\mathbf{x})$ in (1) (e.g., the generalized lasso $\|A\mathbf{x}\|_1$), which is often an

obstacle for applying other methods such as proximal gradient method. As follows, we describe the standard ADMM and its variants for solving (1) in different optimization paradigms. It is worth mentioning that all algorithms presented in this paper can be easily extended to handle a more general term $\psi(\mathcal{A}(\mathbf{x}) + \mathbf{c})$, where $\mathcal{A}$ is a linear mapping.

To apply ADMM, the original problem (1) is first cast into an equivalent constrained optimization problem via decoupling:

$$\min_{\mathbf{x} \in \Omega, \mathbf{y} \in \mathbb{R}^m} \quad f(\mathbf{x}) + \psi(\mathbf{y}), \quad \text{s.t.} \quad \mathbf{y} = A\mathbf{x}. \tag{2}$$

An augmented Lagrangian function for (2) is defined as

$$L(\mathbf{x}, \mathbf{y}, \lambda) = f(\mathbf{x}) + \psi(\mathbf{y}) - \lambda^\top (A\mathbf{x} - \mathbf{y}) + \frac{\beta}{2} \|A\mathbf{x} - \mathbf{y}\|_2^2, \tag{3}$$

where $\beta$ is a constant called penalty parameter and $\lambda \in \mathbb{R}^m$ is a dual variable. Then, the standard ADMM solves problem (1) by executing the following three steps in each iteration:

$$\mathbf{x}_{\tau+1} = \arg\min_{\mathbf{x} \in \Omega} L(\mathbf{x}, \mathbf{y}_\tau, \lambda_\tau) = \arg\min_{\mathbf{x} \in \Omega} f(\mathbf{x}) + \frac{\beta}{2} \left\| (A\mathbf{x} - \mathbf{y}_\tau) - \frac{1}{\beta} \lambda_\tau \right\|_2^2, \tag{4}$$

$$\mathbf{y}_{\tau+1} = \arg\min_{\mathbf{x} \in \Omega} L(\mathbf{x}_{\tau+1}, \mathbf{y}, \lambda_\tau) = \arg\min_{\mathbf{y} \in \mathbb{R}^m} \psi(\mathbf{y}) + \frac{\beta}{2} \left\| (A\mathbf{x}_{\tau+1} - \mathbf{y}) - \frac{1}{\beta} \lambda_\tau \right\|_2^2, \tag{5}$$

$$\lambda_{\tau+1} = \lambda_\tau - \beta(A\mathbf{x}_{\tau+1} - \mathbf{y}_{\tau+1}). \tag{6}$$

When $A$ is not an identity matrix, solving the subproblem (4) above for $\mathbf{x}_{\tau+1}$ might be difficult. To alleviate the issue, **linearized ADMM** [33, 34, 8] has been proposed, which solves the following problem instead of (4):

$$\mathbf{x}_{\tau+1} = \arg\min_{\mathbf{x} \in \Omega} f(\mathbf{x}) + \frac{\beta}{2} \left\| (A\mathbf{x} - \mathbf{y}_\tau) - \frac{1}{\beta} \lambda_\tau \right\|_2^2 + \frac{1}{2} \|\mathbf{x} - \mathbf{x}_\tau\|_G^2, \tag{7}$$

where $\|\mathbf{x}\|_G = \sqrt{\mathbf{x}^\top G \mathbf{x}}$ and $G \in \mathbb{R}^{d \times d}$ is a positive semi-definite matrix. By setting $G = \gamma I - \beta A^\top A \succeq 0$, the term $\mathbf{x}^\top A^\top A\mathbf{x}$ in (7) vanishes. It has been established that both standard ADMM and linearized ADMM have an $O(1/t)$ convergence rate for solving (2) [8] , where $t$ is the number of iterations. Under a minor condition, this result implies an $O(1/\epsilon)$ iteration complexity for solving the original problem (1) (see Corollary 1).

In addition, we consider ADMM for solving (1) in stochastic optimization with

$$f(\mathbf{x}) = \mathrm{E}_\xi[f(\mathbf{x}; \xi)] \tag{8}$$

where $\xi$ is a random variable. This formulation captures many risk minimization problems in machine learning where $\xi$ denotes a data point sampled from a distribution and $f(\mathbf{x}; \xi)$ denotes a loss function of the model $\mathbf{x}$ on the data $\xi$. It also covers as a special case the empirical loss where $f(\mathbf{x}) = \frac{1}{n} \sum_{i=1}^n f(\mathbf{x}; \xi_i)$ with $n$ is the number of samples. For these problems, computing $f(\mathbf{x})$ itself might be prohibitive (e.g., when $n$ is very large) or even impossible. To address this issue, one usually considers the stochastic optimization paradigm, where it is assumed that $f(\mathbf{x}; \xi)$ and its subgradient $\partial f(\mathbf{x}; \xi)$ can be efficiently computed. To solve the stochastic optimization problem, stochastic ADMM algorithms have been proposed [21, 23], which update $\mathbf{y}_{\tau+1}$ and $\lambda_{\tau+1}$ the same to (5) and (6), respectively, but update $\mathbf{x}_{\tau+1}$ as

$$\mathbf{x}_{\tau+1} = \arg\min_{\mathbf{x} \in \Omega} f(\mathbf{x}_\tau; \xi_\tau) + \partial f(\mathbf{x}_\tau; \xi_\tau)^\top (\mathbf{x} - \mathbf{x}_\tau) + \frac{\beta}{2} \left\| (A\mathbf{x} - \mathbf{y}_\tau) - \frac{1}{\beta} \lambda_\tau \right\|_2^2 + \frac{\|\mathbf{x} - \mathbf{x}_\tau\|_{G_\tau}^2}{\eta_\tau} \tag{9}$$

where $\xi_\tau$ is a random sample, $\eta_\tau$ is a stepsize and $G_\tau = \gamma I - \beta \eta_\tau A^\top A \succeq I$ [23] or $G_\tau = I$ [21]. Other stochastic variants of ADMM for general convex optimization were also proposed in [23, 35]. These work have established an $O(1/\sqrt{t})$ convergence rate of stochastic ADMM for solving (2) with $f(\mathbf{x})$ being (8). Under a minor condition, we can also show that these stochastic ADMM algorithms suffer from a higher iteration complexity of $O(1/\epsilon^2)$ for finding an $\epsilon$-optimal solution to the original problem (1) (see Corollary 3).

Although the variants of ADMM with fast convergence rates have been developed under smoothness, strong convexity and other regularity conditions (e.g., the matrix $A$ has full rank), the best iteration

complexities of deterministic ADMM and stochastic ADMM for general convex optimization remain $O(1/\epsilon)$ and $O(1/\epsilon^2)$, respectively. On the other hand, many studies have reported that the performance of ADMM is very sensitive to the penalty parameter $\beta$. How to address or alleviate this issue has attracted many studies and remains an active topic. In particular, it remains an open question how to quantify the improvement in ADMM's theoretical convergence by using adaptive penalty parameters. Of course, the answer to this question depends on the adaptive scheme being used. Almost all previous works focus on using self-adaptive schemes that update the penalty parameter during the course of optimization according to the historical iterates (e.g., by balancing the primal residual and dual residual). However, there is hitherto no quantifiable improvement in terms of convergence rate or iteration complexity for these self-adaptive schemes.

In this paper, we focus on the design of adaptive penalization for both deterministic and stochastic ADMM and show that, with the proposed adaptive updating scheme on the penalty parameter, the theoretical convergence properties of ADMM can be improved without imposing any smoothness and strong convexity assumptions on the objective function. The key difference between the proposed adaptive scheme and previous self-adaptive schemes is that the proposed penalty parameter is adaptive to an local sharpness property of the objective function, namely the local error bound (see Definition 1). Given the exponent constant $\theta \in (0, 1]$ that characterizes this local sharpness property, we show that the proposed deterministic ADMM enjoys an improved iteration complexity of $\widetilde{O}(1/\epsilon^{1-\theta})^2$ and the proposed stochastic ADMM enjoys an iteration complexity of $\widetilde{O}(1/\epsilon^{2(1-\theta)})$, both of which improve the complexity of their standard counterparts which only use a fixed penalty parameter. To the best of our knowledge, this is the first evidence that an adaptive penalty parameter used in ADMM can lead to provably lower iteration complexities. We call the proposed ADMM algorithms locally adaptive ADMM because of its adaptivity to the problem's local property.

## 2  Related Work

Since there is a tremendous amount of studies on ADMM, the review below mainly focuses on the ADMMs with a variable penalty parameter. A convergence rate of $O(1/t)$ was first shown for both the standard and linearized variants of ADMM [8, 19, 9] on general non-smooth and non-strongly convex problems. Later, smoothness and strong convexity assumptions are introduced to develop faster convergence rates of ADMMs [22, 3, 11, 6]. Stochastic ADMM was considered in [21, 23] with a convergence rate of $O(1/\sqrt{t})$ for general convex problems and $\widetilde{O}(1/t)$ for strongly convex problems. Recently, many variance reduction techniques have been borrowed into stochastic ADMM to achieve improved convergence rates for finite-sum optimization problems where $f(\mathbf{x}) = \frac{1}{n}\sum_{i=1}^{n} f_i(\mathbf{x})$ under the smoothness and strong convexity assumptions [37, 36, 24]. Nevertheless, most of these aforementioned works focus on using a constant penalty parameter.

He et al. [10] analyzed ADMM with self-adaptive penalty parameters. The motivation for their self-adaptive penalty is to balance the order of the primal residual and the dual residual. However, the convergence of ADMM with self-adaptive penalty is not guaranteed unless the adaptive scheme is turned off after a number of iterations. Additionally, their self-adaptive rule requires computing the deterministic subgradient of $f(\mathbf{x})$ so that is not appropriate for stochastic optimization. Tian & Yuan [25] proposed a variant of ADMM with variable penalty parameters. Their analysis and algorithm require the smoothness assumption of $\psi(A\mathbf{x})$ and full column rank of the $A$ matrix. Zhou et al. [15] focused on solving low-rank representation by linearized ADMM and also proposed a non-decreasing self-adaptive penalty scheme. However, their scheme is only applicable to an equality constraint $A\mathbf{x} + B\mathbf{y} = \mathbf{c}$ with $\mathbf{c} \neq 0$. Recently, Xu et al. [31] proposed a self-adaptive penalty scheme for ADMM based on the Barzilai and Borwein gradient methods. The convergence of their ADMM relies on the analysis in He et al. [10] and thus requires the penalty parameter to be fixed after a number of iterations. In contrast, our adaptive scheme fpr the penalty parameter is different from the previous methods in the following aspects: (i) it is adaptive to the local sharpness property of the problem; (ii) it allows the penalty parameter to increase to infinity as the algorithm proceeds; (iii) it can be employed for both deterministic and stochastic ADMMs as well as their linearized versions.

It is also notable that the presented algorithms and their convergence theory share many similarities with the recent developments leveraging the local error bound condition [32, 30, 29], where similar iteration complexities have been established. However, we would like to emphasize that the newly

proposed ADMM algorithms are more effective to tackle problems with structured regularizers (e.g., generalized lasso) than the methods in [32, 30, 29], and have an additional unique feature of using adaptive penalty parameter.

## 3 Preliminaries

Recall that the problem of our interest:

$$\min_{\mathbf{x} \in \Omega} F(\mathbf{x}) \triangleq f(\mathbf{x}) + \psi(A\mathbf{x}), \tag{10}$$

where $\Omega \subseteq \mathbb{R}^d$ is a closed convex set, $f : \mathbb{R}^d \to (-\infty, +\infty]$ and $\psi : \mathbb{R}^m \to (-\infty, +\infty]$ are proper lower-semicontinuous convex functions, and $A \in \mathbb{R}^{m \times d}$ is a matrix. Let $\Omega_*$ and $F_*$ denote the optimal set of (10) and the optimal value, respectively. We present some assumptions that will be used in the paper.

**Assumption 1.** *For the convex optimization problem (10), we assume (a) there exist known $\mathbf{x}_0 \in \Omega$ and $\epsilon_0 \geq 0$ such that $F(\mathbf{x}_0) - F_* \leq \epsilon_0$; (b) $\Omega_*$ is a non-empty convex compact set; (c) there exists a constant $\rho$ such that $\|\partial\psi(\mathbf{y})\|_2 \leq \rho$ for all $\mathbf{y}$; (d) $\psi$ is defined everywhere.*

For a positive semi-definite matrix $G$, the $G$-norm is defined as $\|\mathbf{x}\|_G = \sqrt{\mathbf{x}^\top G \mathbf{x}}$. Let $\mathcal{B}(\mathbf{x}, r) = \{\mathbf{u} \in \mathbb{R}^d : \|\mathbf{u} - \mathbf{x}\|_2 \leq r\}$ denote the Euclidean ball centered $\mathbf{x}$ with a radius $r$. We denote by $dist(\mathbf{x}, \Omega_*)$ the distance between $\mathbf{x}$ and the set $\Omega_*$, i.e., $dist(\mathbf{x}, \Omega_*) = \min_{\mathbf{v} \in \Omega_*} \|\mathbf{x} - \mathbf{v}\|_2$. We denote by $\mathcal{S}_\epsilon$ the $\epsilon$-sublevel set of $F(\mathbf{x})$, respectively, i.e., $\mathcal{S}_\epsilon = \{\mathbf{x} \in \Omega : F(\mathbf{x}) \leq F_* + \epsilon\}$.

**Local Sharpness.** Below, we introduce a condition, namely local error bound condition, to characterize the local sharpness property of the objective function.

**Definition 1** (Local error bound (LEB)). *A function $F(\mathbf{x})$ is said to satisfy a local error bound condition on the $\epsilon$-sublevel set if there exist $\theta \in (0, 1]$ and $c > 0$ such that for any $\mathbf{x} \in \mathcal{S}_\epsilon$*

$$dist(\mathbf{x}, \Omega_*) \leq c(F(\mathbf{x}) - F_*)^\theta. \tag{11}$$

**Remark:** We will refer to $\theta$ as the local sharpness parameter. A recent study [1] has shown that the local error bound condition is equivalent to the famous Kurdyka - Łojasiewicz (KL) property [13], which characterizes that under a transformation of $\psi(s) = cs^\theta$, the function $F(\mathbf{x})$ can be made sharp around the optimal solutions, i.e, the norm of subgradient of the transformed function $\psi(F(\mathbf{x}) - F_*)$ is lowered bounded by a constant 1. Note that by allowing $\theta = 0$ in the above condition we can capture a full spectrum of functions. However, a broad family of functions can have a sharper upper bound, i.e., with a non-zero constant $\theta$ in the above condition. For example, for functions that are semi-algebraic and continuous, the above inequality is known to hold on any compact set (c.f. [1] and references therein). The value of $\theta$ has been revealed for many functions (c.f. [18, 14, 20, 1, 32]).

## 4 Locally Adaptive ADMM for Deterministic Optimization

Since the proposed locally adaptive ADMM algorithm builds upon the standard ADMM, we first present the detailed steps of ADMM in Algorithm 1. Note that if we set $G = 0 \in \mathbb{R}^{d \times d}$, it gives the standard ADMM; and if we use $G = \gamma I - \beta A^\top A \succeq 0$, it gives the linearized variant, which can make the computation of $\mathbf{x}_{\tau+1}$ easier. To ensure $G \succeq 0$, the minimum valid value for $\gamma$ in the linearized variant is $\beta\|A\|_2^2$. To present the convergence result of ADMM (Algorithm 1), we first introduce some notations.

$$\mathbf{u} = \begin{pmatrix} \mathbf{x} \\ \mathbf{y} \\ \lambda \end{pmatrix}, \quad \mathcal{F}(\mathbf{u}) = \begin{pmatrix} -A^\top \lambda \\ \lambda \\ A\mathbf{x} - \mathbf{y} \end{pmatrix},$$

$$\widehat{\mathbf{u}}_t = \frac{1}{t}\sum_{\tau=1}^t \mathbf{u}_\tau, \quad \widehat{\mathbf{x}}_t = \frac{1}{t}\sum_{\tau=1}^t \mathbf{x}_\tau, \quad \widehat{\mathbf{y}}_t = \frac{1}{t}\sum_{\tau=1}^t \mathbf{y}_\tau, \quad \widehat{\lambda}_t = \frac{1}{t}\sum_{\tau=1}^t \lambda_\tau.$$

We recall the convergence result of [8] for the equality constrained problem (2), which does not assume any smoothness, strong convexity and other regularity conditions.

| **Algorithm 1** ADMM($\mathbf{x}_0, \beta, t$) | **Algorithm 2** LA-ADMM ($\mathbf{x}_0, \beta_1, K, t$) |
|---|---|
| 1: **Input**: $\mathbf{x}_0 \in \Omega$, the penalty parameter $\beta$, the number of iterations $t$ | 1: **Input**: $\mathbf{x}_0 \in \Omega$, the number of stages $K$, and the number of iterations $t$ per stage, initial value of penalization parameter $\beta_1$ |
| 2: **Initialize:** $\mathbf{x}_1 = \mathbf{x}_0, \mathbf{y}_1 = A\mathbf{x}_1, \lambda_1 = 0, \gamma = \beta\|A\|_2^2$ and $G = \gamma I - \beta A^\top A$ or $G = 0$. | 2: **for** $k = 1, \dots, K$ **do** |
| 3: **for** $\tau = 1, \dots, t$ **do** | 3: $\quad$ Let $\mathbf{x}_k = \text{ADMM}(\mathbf{x}_{k-1}, \beta_k, t)$ |
| 4: $\quad$ Update $\mathbf{x}_{\tau+1}$ by (7), $\mathbf{y}_{\tau+1}$ by (5), | 4: $\quad$ Update $\beta_{k+1} = 2\beta_k$ |
| 5: $\quad$ Update $\lambda_{\tau+1}$ by (6) | 5: **end for** |
| 6: **end for** | 6: **Output**: $\mathbf{x}_K$ |
| 7: **Output**: $\widehat{\mathbf{x}}_t = \sum_{\tau=1}^{t} \mathbf{x}_\tau / t$ | |

**Proposition 1** (Theorem 4.1 in [8]). *For any $\mathbf{x} \in \Omega$, $\mathbf{y} \in \mathbb{R}^m$ and $\lambda \in \mathbb{R}^m$, we have*

$$f(\widehat{\mathbf{x}}_t) + \psi(\widehat{\mathbf{y}}_t) - [f(\mathbf{x}) + \psi(\mathbf{y})] + (\widehat{\mathbf{u}}_t - \mathbf{u})^\top \mathcal{F}(\mathbf{u}) \leq \frac{\|\mathbf{x} - \mathbf{x}_1\|_G^2}{2t} + \frac{\beta\|\mathbf{y} - \mathbf{y}_1\|_2^2}{2t} + \frac{\|\lambda - \lambda_1\|_2^2}{2\beta t}.$$

**Remark:** The above result establishes a convergence rate for the variational inequality pertained to (2). When $t \to \infty$, $(\widehat{\mathbf{x}}_t, \widehat{\mathbf{y}}_t)$ converges to the optimal solutions of (2) in a rate of $O(1/t)$.

Since our goal is to solve the problem (1), next we present a corollary exhibiting the convergence of ADMM for solving the original problem (1). All omitted proofs can be found in the supplement.

**Corollary 1.** *Suppose Assumption 1.c and 1.d hold. Let $\widehat{\mathbf{x}}_t$ be the output of ADMM. For any $\mathbf{x} \in \Omega$, we have*

$$F(\widehat{\mathbf{x}}_t) - F(\mathbf{x}) \leq \frac{\|\mathbf{x} - \mathbf{x}_0\|_G^2}{2t} + \frac{\beta\|A\|_2^2\|\mathbf{x} - \mathbf{x}_0\|_2^2}{2t} + \frac{\rho^2}{2\beta t}.$$

**Remark:** For the standard ADMM with $G = 0$ the first term in the R.H.S vanishes. For the linearized ADMM with $G = \gamma I - \beta A^\top A \succeq 0$, we can bound $\|\mathbf{x} - \mathbf{x}_0\|_G^2 \leq \gamma\|\mathbf{x} - \mathbf{x}_0\|_2^2$. One can also derive a theoretically optimal value of $\beta$ by setting $\mathbf{x} = \mathbf{x}_* \in \Omega_*$ and minimizing the upper bound, which results in $\beta = \frac{\rho}{\|A\|_2\|\mathbf{x}_* - \mathbf{x}_0\|_2}$ for the standard ADMM or $\beta = \frac{\rho}{\sqrt{2}\|A\|_2\|\mathbf{x}_* - \mathbf{x}_0\|_2}$ for the linearized ADMM. Finally, the above result implies that the iteration complexity of standard and linearized ADMM for finding an $\epsilon$-optimal solution of (1) is $O\left(\frac{\rho\|A\|_2\|\mathbf{x} - \mathbf{x}_0\|_2}{\epsilon}\right)$.

Next, we present our locally adaptive ADMM and our main result in this section regarding its iteration complexity. The proposed algorithm is described in Algorithm 2, which is referred to as LA-ADMM. The algorithm runs with multiple stages by calling ADMM at each stage with a warm start and a constant number of iterations $t$. The penalty parameter $\beta_k$ is increased by a constant factor larger than 1 (e.g., 2) after each stage and has an initial value dependent on $\rho$, $\|A\|_2$, $\epsilon_0$, $\theta$ and the targeted accuracy $\epsilon$. The convergence result of LA-ADMM employing $G = \gamma I - \beta A^\top A$ is established below. A slightly better result in terms of a constant factor can be established for employing $G = 0$.

**Theorem 2.** *Suppose Assumption 1 holds and $F(\mathbf{x})$ obeys a local error bound condition on the $\epsilon$-sublevel. Let $\beta_1 = \frac{2\rho\epsilon^{1-\theta}}{\|A\|_2\epsilon_0}$, $K = \lceil \log_2(\epsilon_0/\epsilon) \rceil$ and $t = \left\lceil \frac{8\rho\|A\|_2 \max(1,c^2)}{\epsilon^{1-\theta}} \right\rceil$, we have $F(\mathbf{x}_K) - F_* \leq 2\epsilon$. The iteration complexity of LA-ADMM for achieving an $2\epsilon$-optimal solution is $\widetilde{O}(1/\epsilon^{1-\theta})$.*

**Remark:** There are two levels of adaptivity to the local sharpness of the penalty parameter. First, the initial value $\beta_1$ in Algorithm 3 depends on the local sharpness parameter $\theta$. Second, the time interval to increase the penalty parameter is determined by the value of $t$ which is also dependent on $\theta$. Compared to the iteration complexity $O(1/\epsilon)$ of vanilla ADMM, LA-ADMM can enjoy a lower iteration complexity.

## 5   Locally Adaptive ADMM for Stochastic Optimization

In this section, we consider stochastic optimization problem as the following:

$$\min_{\mathbf{x} \in \Omega} F(\mathbf{x}) \triangleq \mathrm{E}_\xi[f(\mathbf{x}; \xi)] + \psi(A\mathbf{x}), \tag{12}$$

where $\xi$ is a random variable and $f(\mathbf{x}; \xi) : \mathbb{R}^d \to (-\infty, +\infty]$ is a proper lower-semicontinuous convex function for each realization of $\xi$. For this problem, in addition to Assumption 1, we make

| **Algorithm 3** SADMM($\mathbf{x}_0, \eta, \beta, t, \Omega$ ) | **Algorithm 4** LA-SADMM ($\mathbf{x}_0, \eta_1, \beta_1, D_1, K, t$) |
|---|---|
| 1: **Input**: $\mathbf{x}_0 \in \mathbb{R}^d$, a step size $\eta$, penalty parameter $\beta$, the number of iterations $t$ and a domain $\Omega$. | 1: **Input**: $\mathbf{x}_0 \in \mathbb{R}^d$, the number of stages $K$, the number of iterations $t$ per stage, the initial step size $\eta_1$, the initial parameter $\beta_1$ and the initial radius $D_1$. |
| 2: **Initialize**: $\mathbf{x}_1 = \mathbf{x}_0, \mathbf{y}_1 = A\mathbf{x}_1, \lambda_1 = 0$ | 2: **for** $k = 1, \dots, K$ **do** |
| 3: **for** $\tau = 1, \dots, t$ **do** | 3:     Let $\mathbf{x}_k = \text{SADMM}(\mathbf{x}_{k-1}, \eta_k, \beta_k, t, \mathcal{B}_k \cap \Omega)$ |
| 4:     Update $\mathbf{x}_{\tau+1}$ by (9) and $\mathbf{y}_{\tau+1}$ by (5) | 4:     Update $\eta_{k+1} = \eta_k/2$ and $\beta_{k+1} = 2\beta_k$, $D_{k+1} = D_k/2$. |
| 5:     Update $\lambda_{\tau+1}$ by (6) | |
| 6: **end for** | 5: **end for** |
| 7: **Output**: $\widehat{\mathbf{x}}_t = \sum_{\tau=1}^{t} \mathbf{x}_\tau / t$ | 6: **Output**: $\mathbf{x}_K$ |

the following assumption for our development, which is a standard assumption for many previous stochastic gradient methods.

**Assumption 2.** *For the stochastic optimization problem (12), we assume that there exists a constant $R$ such that $\|\partial f(\mathbf{x}; \xi)\|_2 \leq R$ almost surely for any $\mathbf{x} \in \Omega$.*

We present a framework of stochastic ADMM (SADMM) in Algorithm 3. The convergence results for solving the equivalent constrained optimization problem of stochastic ADMM with different choices of $G_\tau$ have been established in [21, 23, 35].

Below, we will focus on $G_\tau = \gamma I - \eta\beta A^\top A \succeq I$ because it leads to computationally more efficient update for $\mathbf{x}_{\tau+1}$ than other two choices for high-dimensional problems. Using $G_\tau = I$ will yield a similar convergence bound except for a constant term and using the idea of AdaGrad for computing $G_\tau$ will lead to the same order of convergence in the worst-case, which we will postpone to future work for exploration. The corollary below will be used in our analysis.

**Corollary 3.** *Suppose Assumption 1.c, 1.d and Assumption 2 hold. Let $G_\tau = \gamma I - \eta\beta A^\top A \succeq I$ in Algorithm 3. For any $\mathbf{x} \in \Omega$,*

$$F(\widehat{\mathbf{x}}_t) - F(\mathbf{x}) \leq \frac{\eta R^2}{2} + \frac{\gamma \|\mathbf{x}_1 - \mathbf{x}\|_2^2}{2\eta t} + \left( \frac{\beta \|A\|_2^2 \|\mathbf{x}_1 - \mathbf{x}\|_2^2}{2t} + \frac{\rho^2}{2\beta t} \right) + \frac{\rho \|A\|_2 \|\mathbf{x}_1 - \mathbf{x}_{t+1}\|_2}{t}$$
$$+ \frac{1}{t} \sum_{\tau=1}^{t} (\mathrm{E}[\mathbf{g}_\tau] - \mathbf{g}_\tau)^\top (\mathbf{x}_\tau - \mathbf{x}).$$

**Remark:** Taking expectation on both sides will yield the expectational convergence bound. We can also use an analysis of large deviation to bound the last term to obtain the convergence with high probability. In particular, by setting $\eta \propto 1/\sqrt{\tau}$, the above result implies an $O(1/\sqrt{t})$ convergence rate, i.e., $O(1/\epsilon^2)$ iteration complexity of stochastic ADMM.

Next, we discuss our locally adaptive stochastic ADMM (LA-SADMM) algorithm in Algorithm 4. The key idea is similar to LA-ADMM, i.e., calling SADMM in multiple stages with warm start. The step size $\eta_k$ in each call of SADMM is fixed and decreases by a certain fraction after one stage. The penalty parameter is updated similarly to that in LA-ADMM but with a different initial value. A key difference from LA-ADMM is that we employ a domain shrinking approach to modify the domain of the solutions $\mathbf{x}_{\tau+1}$ at each stage. For the $k$-th stage, the domain for $\mathbf{x}$ is the intersection of $\Omega$ and $\mathcal{B}_k = \mathcal{B}(\mathbf{x}_{k-1}, D_k)$, where the latter is a ball with a radius of $D_k$ centered at $\mathbf{x}_{k-1}$ (the initial solution of the $k$-th stage). The radius $D_k$ will decrease geometrically between stages. The purpose of using the domain shrinking approach is to tackle the last term of the upper bound in Corollary 3 so that it can decrease geometrically as the stage number increases. A similar idea has been adopted in [29, 7, 5]. Note that during each SADMM, we can use the three choices of $G_\tau$ as mentioned before. Below we only present the convergence result of the variant with $G_\tau = \gamma I - \eta_k\beta_k A^\top A$.

**Theorem 4.** *Suppose Assumptions 1 and 2 hold and $F(\mathbf{x})$ obeys the local error bound condition on $\mathcal{S}_\epsilon$. Given $\delta \in (0,1)$, let $\tilde{\delta} = \delta/K$, $K = \lceil \log_2(\frac{\epsilon_0}{\epsilon}) \rceil$, $\eta_1 = \frac{\epsilon_0}{6R^2}$, $\beta_1 = \frac{6R^2}{\|A\|_2^2 \epsilon_0}$, $D_1 \geq \frac{c\epsilon_0}{\epsilon^{1-\theta}}$, $t$ be the smallest integer such that $t \geq \max\{\frac{6912R^2 \log(1/\tilde{\delta}) D_1^2}{\epsilon_0^2}, \frac{12\rho\|A\|_2 D_1}{\epsilon_0}, \frac{\rho^2 \|A\|_2^2}{R^2}\}$ and $G_\tau = 2I - \eta_1\beta_1 A^\top A \succeq I$. Then LA-SADMM guarantees that, with a probability $1 - \delta$, we have $F(\mathbf{x}_K) - F_* \leq 2\epsilon$. The iteration complexity of LA-SADMM for achieving an $2\epsilon$-optimal solution with a high probability $1 - \delta$ is $\widetilde{O}(\log(1/\delta)/\epsilon^{2(1-\theta)})$, provided $D_1 = O(\frac{c\epsilon_0}{\epsilon^{(1-\theta)}})$.*

| **Algorithm 5** LA-ADMM with Restarting | **Algorithm 6** LA-SADMM with Restarting |
|---|---|
| 1: **Input**: $t_1, \beta_1^{(1)}$ | 1: **Input**: $t_1, D_1^{(1)}$ and $\epsilon \le \epsilon_0/2$ |
| 2: **Initialization**: $\mathbf{x}^{(0)}$ | 2: **Initialization**: $\mathbf{x}^{(0)}, \eta_1 = \frac{\epsilon_0}{6R^2}, \beta_1 = \frac{6R^2}{\|A\|_2^2 \epsilon_0}$ |
| 3: **for** $s = 1, 2, \ldots,$ **do** | 3: **for** $s = 1, 2, \ldots,$ **do** |
| 4:     $\mathbf{x}^{(s)} =$LA-ADMM$(\mathbf{x}^{(s-1)}, \beta_1^{(s)}, K, t_s)$ | 4:     $\mathbf{x}^{(s)} =$LA-SADMM$(\mathbf{x}^{(s-1)}, \eta_1, \beta_1, D_1^{(s)}, K, t_s)$ |
| 5:     $t_{s+1} = t_s 2^{1-\theta}, \beta_1^{(s+1)} = \beta_1^{(s)}/2^{1-\theta}$ | 5:     $t_{s+1} = t_s 2^{2(1-\theta)}, D_1^{(s+1)} = D_1^{(s)} 2^{1-\theta}$ |
| 6: **end for** | 6: **end for** |
| 7: **Output**: $\mathbf{x}^{(S)}$ | 7: **Output**: $\mathbf{x}^{(S)}$ |

**Remark:** Interestingly, unlike that in LA-ADMM, the initial value $\beta_1$ does not depend on $\theta$. The adaptivity of the penalty parameters lies on the time interval $t$ which determines when the value of $\beta$ is increased. The difference comes from the first two terms in Corollary 3.

Before ending this section, we discuss two points. First, both Theorem 2 and Theorem 4 exhibit the dependence of the two algorithms on the $c$ parameter (e.g., $t$ in Algorithm 2 and $D_1$ in Algorithm 4) that is usually unknown. Nevertheless, this issue can be easily addressed by using another level of restarting and increasing sequence of $t$ and $D_1$ similar to the practical variants in [29, 32]. Due to the limit of space, we only present the algorithms in Algorithm 5 and Algorithm 6 with their formal guarantee presented in supplement. The conclusion is that under mild conditions as long as $\beta^{(1)}$ in Algorithm 5 is sufficiently small, $t_1$ and $D_1^{(1)}$ in Algorithm 6 are sufficiently large, the iteration complexities remain $\widetilde{O}(1/\epsilon^{1-\theta})$ and $\widetilde{O}(1/\epsilon^{2(1-\theta)})$ when $\theta$ in LEB condition is known. Second, these variants can be even employed when the local sharpness parameter $\theta$ is unknown by simply setting it to 0, and still enjoy reduced iteration complexities in terms of a multiplicative factor compared to vanilla ADMMs. Detailed results are included in the supplement.

## 6   Applications and Experiments

In this section, we present some experimental results of the proposed algorithms for solving three tasks, namely generalized LASSO, robust regression with a low-rank regularizer (RR-LR) and learning low-rank representation. For generalized lasso, our experiment focuses on comparing the proposed LA-SADMM with SADMM. For the latter tasks, we focus on comparing the proposed LA-ADMM with previous linearized ADMM with and without self-adaptive penalty parameters.

We first consider generalized LASSO, which can find applications in many problems in statistics and machine learning [28]. The objective of generalized LASSO can be expressed as:

$$\min_{\mathbf{x} \in \mathbb{R}^d} F(\mathbf{x}) = \frac{1}{n} \sum_{i=1}^{n} \ell(\mathbf{x}^\top \mathbf{a}_i, b_i) + \delta \|A\mathbf{x}\|_1 \tag{13}$$

where $(\mathbf{a}_i, b_i)$ is a set of pairs of training data, $i = 1, \ldots, n$, $\delta \ge 0$ is a regularization parameter, $A \in \mathbb{R}^{m \times d}$ is a specified matrix, and $\ell(z, b)$ is a convex loss function in terms of $z$. The above formulation include many formulations as special cases, e.g., the standard LASSO where $A = I \in \mathbb{R}^{d \times d}$ [26], fused LASSO that penalizes the $\ell_1$ norm of both the coefficients and their successive differences [27], graph-guided fused LASSO (GGLASSO) where $A = F \in \mathbb{R}^{m \times d}$ encodes some graph information about features [12], and sparse graph-guided fused LASSO (S-GGLASSO) where $\|A\mathbf{x}\|_1 = \delta_2 \|\mathbf{x}\|_1 + \delta_1 \|F\mathbf{x}\|_1$ [21].

Let us first discuss the local sharpness parameter of generalized lasso with different loss functions. For the loss function, let us first consider piecewise linear loss function such as hinge loss $\ell(z, b) = \max(0, 1 - bz)$, absolute loss $\ell(z, b) = |z - b|$ and $\epsilon$-insensitive loss $\ell(z, b) = \max(|z - b| - \epsilon, 0)$. Then the objective is a polyhedral function. According to the results in [32], the local sharpness parameter is $\theta = 1$. It then implies that both LA-ADMM and LA-SADMM enjoy linear convergence results for solving the problem (13) with a piecewise linear loss function. To the best of our knowledge, these are the first linear convergence results of ADMM without smoothness and strong convexity conditions. One can also consider piecewise quadratic loss such as square loss $\ell(z, b) = (z - b)^2$ for $b \in \mathbb{R}$ and squared hinge loss $\ell(z, b) = \max(0, 1 - bz)^2$ for $b \in \{1, -1\}$. According to [14], the problem with convex piecewise quadratic loss has a local sharpness parameter $\theta = 1/2$, implying $\widetilde{O}(1/\sqrt{\epsilon})$ and $\widetilde{O}(1/\epsilon)$ for LA-ADMM and LA-SADMM.

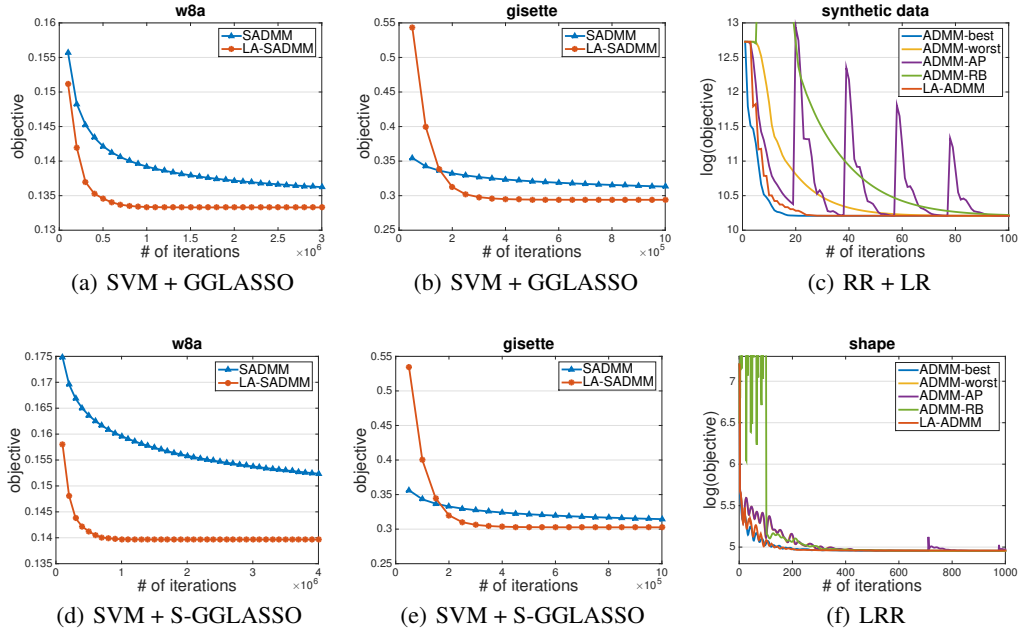

| (a) SVM + GGLASSO | (b) SVM + GGLASSO | (c) RR + LR |
| --- | --- | --- |
| (d) SVM + S-GGLASSO | (e) SVM + S-GGLASSO | (f) LRR |

Figure 1: Comparison of different algorithms for solving different tasks. RR + LR represents robust regression with a low rank regularizer. LRR represents low-rank representation.

For more examples with different values of $\theta$, we refer readers to [32, 30, 29, 17].

**SVM Classification with GGLASSO and S-GGLASSO Regularizers** To generate the $A$ matrix, we first need to construct a dependency graph of features. We follow [21] to generate a dependency graph by sparse inverse covariance selection [4]. Specifically, we get the estimator of the inverse covariance matrix denoted by $\hat{\Sigma}^{-1}$ via sparse inverse covariance estimation with the graphical lasso [4]. For each nonzero entry $\hat{\Sigma}^{-1}_{ij}$, where $i, j \in \{1, \ldots, d\}, i \neq j$, an edge between $i$ and $j$ is created. If we denote by $\mathcal{G} \equiv \{\mathcal{V}, \mathcal{E}\}$ the resulting graph, where $\mathcal{V}$ is a set of $d$ vertices, which correspond to $d$ features in the data, and $\mathcal{E} = \{e_1, \ldots, e_m\}$ denotes the set of $m$ edges between elements of $\mathcal{V}$, where $e_i$ consists of a tuple of two elements, then the $k$-th row of $A$ has two non-zero elements corresponding to the $k$-th edge $e_k = (i, j) \in \mathcal{E}$ with $A_{k,i} = 1$ and $A_{k,j} = -1$. We choose two medium-scale data sets from libsvm website, namely w8a data ($n = 49749, d = 300$) and gisette data ($n = 6000, d = 5000$), to conduct the experiment. In the process of estimating inverse covariance matrix, we choose a penalty parameter to be $0.01$ that renders the percentage of non-zero elements of the $A$ matrix to be around $3\%$ for w8a data and $1\%$ for gisette data. We compare the performance of the LA-SADMM algorithm with SADMM [23], where in SADMM we use $G_\tau = \gamma I - \beta \eta_\tau A^\top A \succeq I$ with $\eta_\tau \propto \eta_1 / \sqrt{\tau}$. For fairness, we set the same initial solution with all zero entries. We fix the value of regularization parameters ($\delta$ in GGLASSO and $\delta_1, \delta_2$ in S-GGLASSO) to be $\frac{1}{n}$, where $n$ is the number of samples. For SADMM, we tune both $\eta_1$ and $\beta$ from $\{10^{-5:1:5}\}$. For LA-SADMM, we set the initial step size and penalty parameter to their theoretical value in Theorem 4, and select $D_1$ from $\{100, 1000\}$. The values of $t$ in LA-SADMM is set to $10^5$ and $5 \times 10^4$ for w8a and gisette, respectively. The results of comparing the objective values versus the number of iterations are presented in Figure 1 (a,b,d,e). We can see that LA-SADMM exhibits a much faster convergence than SADMM.

**Robust Regression with a Low-rank Regularizer** The objective function is $F(X) = \lambda \|X\|_* + \|AX - C\|_1$. We can form an equality constraint $Y = AX - C$ and solve the problem by linearized ADMM. The value of the local sharpness parameter of this problem is still an open problem. We compare the proposed LA-ADMM, the vanilla linearized ADMM with a fixed penalty parameter (ADMM), the linearized ADMM with self-adaptive penalty proposed in [15] (ADMM-AP), and the linearized ADMM with residual balancing in [10, 2] (ADMM-RB). We construct a synthetic data where $A \in \mathbb{R}^{1000 \times 100}$ is generated following a Gaussian distribution with mean 0 and standard deviation 1. To construct $C \in \mathbb{R}^{1000 \times 50}$, we first generate $X \in \mathbb{R}^{100 \times 50}$ and

retain only its top 20 components denoted by $\hat{X}$ and then let $C = A\hat{X} + \varepsilon$, where $\varepsilon$ is a Gaussian noise matrix with mean zero and standard deviation 0.01. We set $\lambda = 100$. For the vanilla linearized ADMM, we try different penalty parameters from $\{10^{-3:1:3}\}$ and report the best performance (using $\beta = 0.01$) and worst performance (using $\beta = 0.001$). To demonstrate the capability of adaptive ADMM, we choose $\beta = 0.001$ as the initial step size for LA-ADMM and ADMM-AP. Other parameters of ADMM-AP is the same as suggested in the original paper. For LA-ADMM, we implement its restarting variant (Algorithm 5), and start with the number of inner iterations $t = 2$ and increase its value by a factor 2 after 10 stages, and also increase the value of $\beta$ by 10 times after each stage. The results are reported in Figure 1 (c), from which we can see that LA-ADMM performs comparably with ADMM with the best penalty parameter and also better than ADMM-AP. We also include the results in terms of running time in the supplement.

**Low-rank Representation [16]** The objective function is $F(X) = \lambda\|X\|_* + \|AX - A\|_{2,1}$, where $A \in \mathbb{R}^{n \times d}$ is a data matrix. We used the shape image [3] and set $\lambda = 10$. For the vanilla linearized ADMM, we try different penalty parameters from $\{10^{-3:1:3}\}$ and report the best performance (using $\beta = 0.1$) and worst performance (using $\beta = 0.01$). To demonstrate the capability of adaptive ADMM, we choose $\beta = 0.01$ as the initial step size for LA-ADMM and ADMM-AP. Other parameters of ADMM-AP is the same as suggested in the original paper. For LA-ADMM, we start with the number of inner iterations $t = 20$ and increase its value by a factor 2 after 2 stages, and also increase the value of $\beta$ by 2 times after each stage. The results are reported in Figure 1 (f), from which we can see that LA-ADMM performs comparably with ADMM with the best penalty parameter and also better than ADMM-AP. We can see from the figure that the results of ADMM-worst and ADMM-AP are quite similar. We also include the results in terms of running time in the supplement.

## 7 Conclusion

In this paper, we have presented a new theory of (linearized) ADMM for both deterministic and stochastic optimization with adaptive penalty parameters. The new adaptive scheme is different from previous self-adaptive schemes and is adaptive to the local sharpness of the problem. We have established faster convergence of the proposed algorithms of ADMM with penalty parameters adaptive to the local sharpness parameter. Experimental results have demonstrated the superior performance of the proposed stochastic and deterministic adaptive ADMM.

## Acknowlegements

We thank the anonymous reviewers for their helpful comments. Y. Xu, M. Liu and T. Yang are partially supported by National Science Foundation (IIS-1463988, IIS-1545995). Y. Xu would like to thank Yan Yan for useful discussions on the low-rank representation experiments.

## Footnotes

[1]$\widetilde{O}()$ suppresses a logarithmic factor.

[2]$\widetilde{O}()$ suppresses a logarithmic factor.

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
