[Supplementary Material · adaptive-admm-supplement.pdf]

# Supplementary Materials
# "ADMM without a Fixed Penalty Parameter: Faster Convergence with New Adaptive Penalization"

**Yi Xu[†], Mingrui Liu[†], Qihang Lin[‡], Tianbao Yang[†]**
[†]Department of Computer Science, The University of Iowa, Iowa City, IA 52242, USA
[‡]Department of Management Sciences, The University of Iowa, Iowa City, IA 52242, USA
{yi-xu, mingrui-liu, qihang-lin, tianbao-yang}@uiowa.edu

## 1 Proof of Corollary 1

**Corollary 1.** *Suppose Assumption 1.c and 1.d hold. Let $\widehat{\mathbf{x}}_t$ be the output of ADMM. For any $\mathbf{x} \in \Omega$, we have*

$$F(\widehat{\mathbf{x}}_t) - F(\mathbf{x}) \leq \frac{\|\mathbf{x} - \mathbf{x}_0\|_G^2}{2t} + \frac{\beta\|A\|_2^2\|\mathbf{x} - \mathbf{x}_0\|_2^2}{2t} + \frac{\rho^2}{2\beta t}.$$

*Proof.* Let $\mathbf{y} = A\mathbf{x}$, we have

$$
\begin{aligned}
(\widehat{\mathbf{u}}_t - \mathbf{u})^\top \mathcal{F}(\mathbf{u}) &= (\widehat{\mathbf{x}}_t - \mathbf{x})^\top(-A^\top\lambda) + (\widehat{\mathbf{y}}_t - \mathbf{y})^\top\lambda + (\widehat{\lambda}_t - \lambda)^\top(A\mathbf{x} - \mathbf{y}) \\
&= -\lambda^\top(A\widehat{\mathbf{x}}_t - \widehat{\mathbf{y}}_t) + \widehat{\lambda}_t^\top(A\mathbf{x} - \mathbf{y}) \\
&= -\lambda^\top(A\widehat{\mathbf{x}}_t - \widehat{\mathbf{y}}_t).
\end{aligned}
$$

Then following Proposition 1, we have

$$f(\widehat{\mathbf{x}}_t) + \psi(\widehat{\mathbf{y}}_t) - [f(\mathbf{x}) + \psi(A\mathbf{x})] - \lambda^\top(A\widehat{\mathbf{x}}_t - \widehat{\mathbf{y}}_t) \leq \frac{\|\mathbf{x} - \mathbf{x}_1\|_G^2}{2t} + \frac{\beta\|A(\mathbf{x} - \mathbf{x}_1)\|_2^2}{2t} + \frac{\|\lambda - \lambda_1\|_2^2}{2\beta t}.$$

Since the above inequality holds for any $\lambda \in \mathbb{R}^m$, we can maximize both sides over $\|\lambda\|_2 \leq \rho$, and by noting $\lambda_1 = 0$ we have

$$f(\widehat{\mathbf{x}}_t) + \psi(\widehat{\mathbf{y}}_t) - [f(\mathbf{x}) + \psi(A\mathbf{x})] + \rho\|A\widehat{\mathbf{x}}_t - \widehat{\mathbf{y}}_t\|_2 \leq \frac{\|\mathbf{x} - \mathbf{x}_1\|_G^2}{2t} + \frac{\beta\|A(\mathbf{x} - \mathbf{x}_1)\|_2^2}{2t} + \frac{\rho^2}{2\beta t}.$$

By Assumption 1.c, we have

$$\psi(A\widehat{\mathbf{x}}_t) - \psi(\widehat{\mathbf{y}}_t) \leq \rho\|A\widehat{\mathbf{x}}_t - \widehat{\mathbf{y}}_t\|_2.$$

Thus,

$$f(\widehat{\mathbf{x}}_t) + \psi(A\widehat{\mathbf{x}}_t) - [f(\mathbf{x}) + \psi(A\mathbf{x})] \leq \frac{\|\mathbf{x} - \mathbf{x}_1\|_G^2}{2t} + \frac{\beta\|A(\mathbf{x} - \mathbf{x}_1)\|_2^2}{2t} + \frac{\rho^2}{2\beta t}.$$

which completes the proof by noting that $F(\mathbf{x}) = f(\mathbf{x}) + \psi(A\mathbf{x})$.  □

## 2 Proof of Theorem 2

**Theorem 2.** *Suppose Assumption 1 holds and $F(\mathbf{x})$ obeys a local error bound condition on the $\epsilon$-sublevel. Let $\beta_1 = \frac{2\rho\epsilon^{1-\theta}}{\|A\|_2\epsilon_0}$, $K = \lceil \log_2(\epsilon_0/\epsilon) \rceil$ and $t = \left\lceil \frac{8\rho\|A\|_2\max(1,c^2)}{\epsilon^{1-\theta}} \right\rceil$, we have $F(\mathbf{x}_K) - F_* \leq 2\epsilon$. The iteration complexity of LA-ADMM for achieving an $2\epsilon$-optimal solution is $\widetilde{O}(1/\epsilon^{1-\theta})$.*

To prove the theorem, we first present a lemma due to [2].

**Lemma 1.** *[2] For any* $\mathbf{x} \in \Omega$ *and* $\epsilon > 0$, *we have*

$$\|\mathbf{x} - \mathbf{x}_\epsilon^\dagger\|_2 \leq \frac{dist(\mathbf{x}_\epsilon^\dagger, \Omega_*)}{\epsilon}(F(\mathbf{x}) - F(\mathbf{x}_\epsilon^\dagger))$$

*where* $\mathbf{x}_\epsilon^\dagger \in \mathcal{S}_\epsilon$ *is the closest point in the* $\epsilon$*-sublevel set to* $\mathbf{x}$.

*Proof of Theorem 2.* Here we only prove for the case of $G = \gamma I - \beta A^\top A$ with $\gamma = \beta\|A\|_2^2$. Following the same analysis, we can easily prove the same result for using $G = 0$. Let $\mathbf{x}_{k-1,\epsilon}^\dagger$ denote the closest point to $\mathbf{x}_{k-1}$ in $\mathcal{S}_\epsilon$. Define $\epsilon_k = \frac{\epsilon_0}{2^k}$. Then $\beta_k = \frac{\rho\epsilon^{1-\theta}}{\|A\|_2\epsilon_k}$ and $\gamma_k = \beta_k\|A\|_2^2 = \frac{\rho\epsilon^{1-\theta}\|A\|_2}{\epsilon_k}$. We prove this by induction. Assume $F(\mathbf{w}_{k-1}) - F_* \leq \epsilon_{k-1} + \epsilon$, which trivally holds for $k = 1$ due to Assumption 1.a. We apply Corollary 1 to the $k$-th stage of LA-ADMM. For any $\mathbf{x} \in \Omega$, we have

$$F(\mathbf{x}_k) - F(\mathbf{x}) \leq \frac{\gamma_k\|\mathbf{x} - \mathbf{x}_{k-1}\|_2^2}{2t} + \frac{\beta_k\|A\|_2^2\|\mathbf{x} - \mathbf{x}_{k-1}\|_2^2}{2t} + \frac{\rho^2}{2\beta_k t}.$$

Let $\mathbf{x} = \mathbf{x}_{k-1,\epsilon}^\dagger$ so that we have

$$F(\mathbf{x}_k) - F(\mathbf{x}_{k-1,\epsilon}^\dagger) \leq \frac{\gamma_k\|\mathbf{x}_{k-1,\epsilon}^\dagger - \mathbf{x}_{k-1}\|_2^2}{2t} + \frac{\beta_k\|A\|_2^2\|\mathbf{x}_{k-1,\epsilon}^\dagger - \mathbf{x}_{k-1}\|_2^2}{2t} + \frac{\rho^2}{2\beta_k t}. \tag{1}$$

We consider two scenarios of $\mathbf{x}_{k-1}$. First, suppose $\mathbf{x}_{k-1} \in \mathcal{S}_\epsilon$ so that $\mathbf{x}_{k-1} = \mathbf{x}_{k-1,\epsilon}^\dagger$. Then

$$F(\mathbf{x}_k) - F(\mathbf{x}_{k-1,\epsilon}^\dagger) \leq \frac{\rho^2}{2\beta_k t} \leq \frac{\rho^2\|A\|_2\epsilon_k\epsilon^{1-\theta}}{2\rho\epsilon^{1-\theta}8\rho\|A\|_2} = \frac{\epsilon_k}{16},$$

which implies

$$F(\mathbf{x}_k) - F_* \leq F(\mathbf{x}_{k-1,\epsilon}^\dagger) - F_* + \epsilon_k \leq \epsilon + \epsilon_k.$$

Secondly, suppose $\mathbf{x}_{k-1} \notin \mathcal{S}_\epsilon$ so that $F(\mathbf{x}_{k-1,\epsilon}^\dagger) = F_* + \epsilon$. By Lemma 1 and the local error bound condition of $F$, we have

$$\|\mathbf{x}_{k-1} - \mathbf{x}_{k-1,\epsilon}^\dagger\|_2 \leq \frac{dist(\mathbf{x}_{k-1,\epsilon}^\dagger, \Omega_*)}{\epsilon}(F(\mathbf{x}_{k-1}) - F(\mathbf{x}_{k-1,\epsilon}^\dagger)) \leq \frac{c\epsilon^\theta}{\epsilon}\epsilon_{k-1} = \frac{c\epsilon_{k-1}}{\epsilon^{1-\theta}}.$$

where we use the assumption that $F(\mathbf{x}_{k-1}) - F_* \leq \epsilon + \epsilon_{k-1}$ and the fact $F(\mathbf{x}_{k-1,\epsilon}^\dagger) = F_* + \epsilon$. Plugging the above bound into (1) we have

$$F(\mathbf{x}_k) - F(\mathbf{x}_{k-1,\epsilon}^\dagger) \leq \frac{\beta_k\|A\|_2^2c^2\epsilon_{k-1}^2}{2t\epsilon^{2(1-\theta)}} + \frac{\beta_k\|A\|_2^2c^2\epsilon_{k-1}^2}{2t\epsilon^{2(1-\theta)}} + \frac{\rho^2}{2\beta_k t}$$

$$= \frac{\rho\epsilon^{1-\theta}\|A\|_2^2c^2\epsilon_{k-1}^2}{\|A\|_2\epsilon_k t\epsilon^{2(1-\theta)}} + \frac{\rho^2\epsilon_k\|A\|_2}{2\rho\epsilon^{1-\theta}t}.$$

Since $t \geq \frac{8\rho\|A\|_2\max(c^2,1)}{\epsilon^{1-\theta}}$, we have

$$F(\mathbf{x}_k) - F(\mathbf{x}_{k-1,\epsilon}^\dagger) \leq \frac{\epsilon_k}{2} + \frac{\epsilon_k}{16} \leq \epsilon_k,$$

which implies

$$F(\mathbf{x}_k) - F_* \leq \epsilon_k + \epsilon.$$

We can finish the proof by the induction up to $k = K = \lceil\log_2(\epsilon_0/\epsilon)\rceil$, which yields $F(\mathbf{x}_K) - F_* \leq 2\epsilon$. □

# 3 Proof of Corollary 3

**Corollary 3.** *Suppose Assumption 1.c, 1.d and Assumption 2 hold. Let $G_\tau = \gamma I - \eta\beta A^\top A \succeq I$ in Algorithm 3. For any $\mathbf{x} \in \Omega$,*

$$F(\widehat{\mathbf{x}}_t) - F(\mathbf{x}) \leq \frac{\eta R^2}{2} + \frac{\gamma\|\mathbf{x}_1 - \mathbf{x}\|_2^2}{2\eta t} + \left(\frac{\beta\|A\|_2^2\|\mathbf{x}_1 - \mathbf{x}\|_2^2}{2t} + \frac{\rho^2}{2\beta t}\right) + \frac{\rho\|A\|_2\|\mathbf{x}_1 - \mathbf{x}_{t+1}\|_2}{t}$$

$$+ \frac{1}{t}\sum_{\tau=1}^{t}(\mathrm{E}[\mathbf{g}_\tau] - \mathbf{g}_\tau)^\top(\mathbf{x}_\tau - \mathbf{x}).$$

To prove this Corollary, we first present a theorem whose proof will be presented later.

**Theorem 5.** *Suppose Assumption 1.c and 1.d hold. By running Algorithm 3 with $t$ iterations, for any $\mathbf{x} \in \Omega$ we hae*

$$F(\widehat{\mathbf{x}}_t) - F(\mathbf{x}) \leq \frac{\eta}{t}\sum_{\tau=1}^{t}\frac{\|\mathbf{g}_\tau\|_{G_\tau^{-1}}^2}{2} + \frac{1}{\eta t}\sum_{\tau=1}^{t}\left(\frac{\|\mathbf{x}_\tau - \mathbf{x}\|_{G_\tau}^2}{2} - \frac{\|\mathbf{x}_{\tau+1} - \mathbf{x}\|_{G_\tau}^2}{2}\right)$$

$$+ \left(\frac{\beta\|A(\mathbf{x}_1 - \mathbf{x})\|_2^2}{2t} + \frac{\rho^2}{2\beta T}\right) + \frac{\rho\|A(\mathbf{x}_1 - \mathbf{x}_{T+1})\|_2}{t} + \frac{1}{t}\sum_{\tau=1}^{t}(\mathrm{E}[\mathbf{g}_\tau] - \mathbf{g}_\tau)^\top(\mathbf{x}_\tau - \mathbf{x})$$

*Proof of Corollary 3.* By Theorem 5, we have

$$F(\widehat{\mathbf{x}}_t) - F(\mathbf{x}) \leq \frac{\eta}{t}\sum_{\tau=1}^{t}\frac{\|\mathbf{g}_\tau\|_{G_\tau^{-1}}^2}{2} + \frac{1}{\eta t}\sum_{\tau=1}^{t}\left(\frac{\|\mathbf{x}_\tau - \mathbf{x}\|_{G_\tau}^2}{2} - \frac{\|\mathbf{x}_{\tau+1} - \mathbf{x}\|_{G_\tau}^2}{2}\right)$$

$$+ \left(\frac{\beta\|A\|_2^2\|\mathbf{x}_1 - \mathbf{x}\|_2^2}{2t} + \frac{\rho^2}{2\beta t}\right) + \frac{\rho\|A\|_2\|\mathbf{x}_1 - \mathbf{x}_{t+1}\|_2}{t} + \frac{1}{t}\sum_{\tau=1}^{t}(\mathrm{E}[\mathbf{g}_\tau] - \mathbf{g}_\tau)^\top(\mathbf{x}_\tau - \mathbf{x}). \quad (2)$$

Since $G_\tau = G_\tau^\top$ and $G_\tau \succeq I$, we have $G_\tau^{-1} \preceq I$ so that the first term in the R.H.S. of (2) is bounded by

$$\frac{\eta}{t}\sum_{\tau=1}^{t}\frac{\|\mathbf{g}_\tau\|_{G_\tau^{-1}}^2}{2} \leq \frac{\eta}{t}\sum_{\tau=1}^{t}\frac{\|\mathbf{g}_\tau\|_2^2}{2} \leq \frac{\eta}{t}\sum_{\tau=1}^{t}\frac{R^2}{2} = \frac{\eta R^2}{2}. \quad (3)$$

On the other hand,

$$\frac{1}{\eta t}\sum_{\tau=1}^{t}\left(\frac{\|\mathbf{x}_\tau - \mathbf{x}\|_{G_\tau}^2}{2} - \frac{\|\mathbf{x}_{\tau+1} - \mathbf{x}\|_{G_\tau}^2}{2}\right) = \frac{1}{\eta t}\left(\frac{\|\mathbf{x}_1 - \mathbf{x}\|_{G_\tau}^2}{2} - \frac{\|\mathbf{x}_{t+1} - \mathbf{x}\|_{G_\tau}^2}{2}\right)$$

$$\leq \frac{\|\mathbf{x}_1 - \mathbf{x}\|_{G_\tau}^2}{2\eta t} \leq \frac{\gamma\|\mathbf{x}_1 - \mathbf{x}\|_2^2}{2\eta t}. \quad (4)$$

Plugging inequalities (3) and (4) into (2), we complete the proof. $\qquad\square$

# 4 Proof of Theorem 4

**Theorem 4.** *Suppose Assumptions 1 and 2 hold and $F(\mathbf{x})$ obeys the local error bound condition on $\mathcal{S}_\epsilon$. Given $\delta \in (0,1)$, let $\tilde{\delta} = \delta/K$, $K = \lceil\log_2(\frac{\epsilon_0}{\epsilon})\rceil$, $\eta_1 = \frac{\epsilon_0}{6R^2}$, $\beta_1 = \frac{6R^2}{\|A\|_2^2\epsilon_0}$, $D_1 \geq \frac{c\epsilon_0}{\epsilon^{1-\theta}}$, $t$ be the smallest integer such that $t \geq \max\{\frac{6912R^2\log(1/\tilde{\delta})D_1^2}{\epsilon_0^2}, \frac{12\rho\|A\|_2 D_1}{\epsilon_0}, \frac{\rho^2\|A\|_2^2}{R^2}\}$ and $G_\tau = 2I - \eta_1\beta_1 A^\top A \succeq I$. Then LA-SADMM guarantees that, with a probability $1 - \delta$, we have $F(\mathbf{x}_K) - F_* \leq 2\epsilon$. The iteration complexity of LA-SADMM for achieving an $2\epsilon$-optimal solution with a high probability $1 - \delta$ is $\widetilde{O}(\log(1/\delta)/\epsilon^{2(1-\theta)})$ provided $D_1 = O(\frac{c\epsilon_0}{\epsilon^{(1-\theta)}})$.*

To prove Theorem 4, we first bound the last term of the upper bound in Corollary 3 using the following lemma whose proof can be found in [1] (in the proof of their Lemma 10).

**Lemma 2.** *Suppose Assumption 2 holds. Given $\mathbf{x}_{k-1}$, let $\mathbf{x}_{k-1,\epsilon}^{\dagger}$ be the closest solution to $\mathbf{x}_{k-1}$ in the $\epsilon$-sublevel set $\mathcal{S}_\epsilon$. Let $D_k$ be the upper bound of $\|\mathbf{x}_{k-1} - \mathbf{x}_{k-1,\epsilon}^{\dagger}\|_2$. Apply $t$-iterations of $\mathbf{x}_{\tau+1}^k = \Pi_{\Omega \cap \mathcal{B}(\mathbf{x}_{k-1}, D_k)}[\mathbf{x}_\tau^k - \eta \mathbf{g}_\tau^k]$, where $\mathrm{E}[\mathbf{g}_\tau^k] \in \partial f(\mathbf{x}_\tau^k)$. For any $\delta \in (0,1)$, with a probability of at least $1 - \delta$, we have*

$$\frac{1}{t}\sum_{\tau=1}^{t}(\mathrm{E}[\mathbf{g}_\tau^k] - \mathbf{g}_\tau^k)^\top(\mathbf{x}_{k-1,\epsilon}^{\dagger} - \mathbf{x}_\tau^k) \leq \frac{4RD_k\sqrt{3\log(1/\delta)}}{\sqrt{t}}.$$

*Proof of Theorem 4.* To prove the theorem, we apply Corollary 3 to each stage of LA-SADMM with $\mathbf{x} = \mathbf{x}_{k-1,\epsilon}^{\dagger}$, where $\mathbf{x}_{k-1,\epsilon}^{\dagger}$ denotes the closest solution to $\mathbf{x}_{k-1}$ in the $\epsilon$-sublevel set. We will prove this by induction. Define $\epsilon_k = \frac{\epsilon_0}{2^k}$. Let us assume that $F(\mathbf{x}_{k-1}) - F_* \leq \epsilon_{k-1} + \epsilon$. First, we need to show that $\mathbf{x}_{k-1,\epsilon}^{\dagger} \in \Omega \cap \mathcal{B}(\mathbf{x}_{k-1}, D_k)$. It suffices to show $\|\mathbf{x}_{k-1,\epsilon}^{\dagger} - \mathbf{x}_{k-1}\|_2 \leq D_k$, which is true because

$$
\begin{aligned}
\|\mathbf{x}_{k-1,\epsilon}^{\dagger} - \mathbf{x}_{k-1}\|_2 &\leq \frac{dist(\mathbf{x}_{k-1,\epsilon}^{\dagger}, \Omega_*)}{\epsilon}(F(\mathbf{x}_{k-1}) - F(\mathbf{x}_{k-1,\epsilon}^{\dagger})) \\
&\leq \frac{dist(\mathbf{x}_{k-1,\epsilon}^{\dagger}, \Omega_*)}{\epsilon}[\epsilon_{k-1} + \epsilon - \epsilon] = \frac{dist(\mathbf{x}_{k-1,\epsilon}^{\dagger}, \Omega_*)\epsilon_{k-1}}{\epsilon} \\
&\leq \frac{c(F(\mathbf{x}_{k-1,\epsilon}^{\dagger}) - F_*)^\theta \epsilon_{k-1}}{\epsilon} \leq \frac{c\epsilon^\theta \epsilon_{k-1}}{\epsilon} = \frac{c\epsilon_{k-1}}{\epsilon^{1-\theta}} \leq D_k.
\end{aligned}
$$

Then, by Corollary 3 and Lemma 2, with a probability $1 - \tilde{\delta}$, we have

$$
\begin{aligned}
F(\mathbf{x}_k) - F(\mathbf{x}_{k-1,\epsilon}^{\dagger}) &\leq \frac{\eta_k R^2}{2} + \frac{\|\mathbf{x}_{k-1} - \mathbf{x}_{k-1,\epsilon}^{\dagger}\|_2^2}{2\eta_k t} + \left(\frac{\beta_k\|A\|_2^2\|\mathbf{x}_{k-1} - \mathbf{x}_{k-1,\epsilon}^{\dagger}\|_2^2}{2t} + \frac{\rho^2}{2\beta_k t}\right) \\
&\quad + \frac{\rho\|A\|_2\|\mathbf{x}_{k-1} - \mathbf{x}_{t+1}^k\|_2}{t} + \frac{4RD_k\sqrt{3\log(1/\tilde{\delta})}}{\sqrt{t}} \\
&\leq \frac{\eta_k R^2}{2} + \frac{c^2\epsilon_{k-1}^2}{2\eta_k t\epsilon^{2(1-\theta)}} + \frac{\beta_k\|A\|_2^2 c^2\epsilon_{k-1}^2}{2t\epsilon^{2(1-\theta)}} + \frac{\rho^2}{2\beta_k t} \\
&\quad + \frac{\rho\|A\|_2 D_k}{t} + \frac{4RD_k\sqrt{3\log(1/\tilde{\delta})}}{\sqrt{t}}.
\end{aligned}
\tag{5}
$$

where the second inequality is from (5) and the fact that $\mathbf{x}_{t+1}^k \in \mathcal{B}(\mathbf{x}_{k-1}, D_k)$. Setting $\eta_k = \frac{\epsilon_k}{3R^2}$, $\beta_k = \frac{3R^2}{\|A\|_2^2 \epsilon_k}$ and $t \geq \max\left\{\frac{\rho^2\|A\|_2^2}{R^2}, \frac{12\rho\|A\|_2 D_1}{\epsilon_0}, \frac{6912R^2\log(1/\tilde{\delta})D_1^2}{\epsilon_0^2}\right\}$ in the right hand side of the inequality above, we have

$$F(\mathbf{x}_k) - F(\mathbf{x}_{k-1,\epsilon}^{\dagger}) \leq \frac{\epsilon_k}{6} \times 6 = \epsilon_k.$$

Hence, conditioned on $F(\mathbf{x}_{k-1}) - F_* \leq \epsilon_{k-1} + \epsilon$, we have

$$F(\mathbf{w}_k) - F_* \leq \epsilon_k + \epsilon$$

with a probability of $1 - \tilde{\delta}$. By induction, with a probability of $(1 - \tilde{\delta})^K \geq 1 - \delta$, we have

$$F(\mathbf{w}_K) - F_* \leq \epsilon_K + \epsilon \leq 2\epsilon.$$

$\square$

## 5  Proof of Theorem 5

*Proof.* To prove the theorem, we first introduce some notations and technical lemmas. Define

$$\mathbf{u} = \begin{pmatrix} \mathbf{x} \\ \mathbf{y} \\ \lambda \end{pmatrix}, \quad \mathcal{F}(\mathbf{u}) = \begin{pmatrix} -A^\top \lambda \\ \lambda \\ A\mathbf{x} - \mathbf{y} \end{pmatrix},$$

$$\Delta_t = (\partial f(\mathbf{x}_t) - \mathbf{g}_t)^\top(\mathbf{x}_t - \mathbf{x}), \quad \widehat{\mathbf{u}}_T = \frac{1}{T}\sum_{t=1}^{T}\mathbf{u}_{t+1}.$$

**Lemma 3.** *Let $G \succ 0$ and $\mathbf{w}_+$ be given by*

$$\mathbf{w}_+ = \arg\min_{\mathbf{x} \in \Omega} h(\mathbf{x}) + \frac{1}{2\eta}\|\mathbf{x} - \mathbf{w}\|_G^2 \tag{6}$$

*Then for any $\mathbf{x} \in \Omega$*

$$\nabla h(\mathbf{w}_+)^\top (\mathbf{w}_+ - \mathbf{x}) \leq \frac{1}{2\eta}\left(\|\mathbf{w} - \mathbf{x}\|_G^2 - \|\mathbf{w}_+ - \mathbf{x}\|_G^2 - \|\mathbf{w}_+ - \mathbf{w}\|_G^2\right)$$

*Proof.* By the optimality condition, we have

$$(\mathbf{x} - \mathbf{w}_+)^\top \left(\nabla h(\mathbf{w}_+) + \frac{1}{\eta}G(\mathbf{w}_+ - \mathbf{w})\right) \geq 0, \quad \forall x \in \Omega.$$

It is easy to verify that

$$\frac{1}{\eta}(\mathbf{x} - \mathbf{w}_+)^\top G(\mathbf{w}_+ - \mathbf{w}) = \frac{1}{2\eta}\left(\|\mathbf{w} - \mathbf{x}\|_G^2 - \|\mathbf{w}_+ - \mathbf{x}\|_G^2 - \|\mathbf{w}_+ - \mathbf{w}\|_G^2\right).$$

$\square$

We now begin to prove Theorem 5. By the convexity of $f(\mathbf{x})$, for any $x \in \Omega$, we have

$$f(\mathbf{x}_t) - f(\mathbf{x}) \leq \partial f(\mathbf{x}_t)^\top (\mathbf{x}_t - \mathbf{x}) = \mathbf{g}_t^\top (\mathbf{x}_t - \mathbf{x}) + \Delta_t = \mathbf{g}_t^\top (\mathbf{x}_{t+1} - \mathbf{x}) + \mathbf{g}_t^\top (\mathbf{x}_t - \mathbf{x}_{t+1}) + \Delta_t. \tag{7}$$

Applying Lemma 3 to

$$\mathbf{x}_{t+1} = \arg\min_{\mathbf{x} \in \Omega} \mathbf{g}_t^\top \mathbf{x} - \mathbf{x}^\top A^\top \lambda_t + \frac{\beta}{2}\|A\mathbf{x} - \mathbf{y}_t\|_2^2 + \frac{1}{2\eta}\|\mathbf{x} - \mathbf{x}_t\|_{G_t}^2$$

leads to

$$\left(\mathbf{g}_t - A^\top \lambda_t + \beta A^\top (A\mathbf{x}_{t+1} - \mathbf{y}_t)\right)^\top (\mathbf{x}_{t+1} - \mathbf{x}) \leq \frac{1}{2\eta}\left(\|\mathbf{x}_t - \mathbf{x}\|_{G_t}^2 - \|\mathbf{x}_{t+1} - \mathbf{x}\|_{G_t}^2 - \|\mathbf{x}_{t+1} - \mathbf{x}_t\|_{G_t}^2\right).$$

which further implies

$$\mathbf{g}_t^\top (\mathbf{x}_{t+1} - \mathbf{x}) \leq \frac{1}{2\eta}\left(\|\mathbf{x}_t - \mathbf{x}\|_{G_t}^2 - \|\mathbf{x}_{t+1} - \mathbf{x}\|_{G_t}^2 - \|\mathbf{x}_{t+1} - \mathbf{x}_t\|_{G_t}^2\right) - (\mathbf{x}_{t+1} - \mathbf{x})^\top A^\top (\beta(A\mathbf{x}_{t+1} - \mathbf{y}_t) - \lambda_t).$$

Then, combining the inequality above with (7), we have

$$f(\mathbf{x}_t) - f(\mathbf{x}) - (\mathbf{x}_{t+1} - \mathbf{x})^\top A^\top \lambda_{t+1}$$

$$\leq \frac{1}{2\eta}\left(\|\mathbf{x}_t - \mathbf{x}\|_{G_t}^2 - \|\mathbf{x}_{t+1} - \mathbf{x}\|_{G_t}^2 - \|\mathbf{x}_{t+1} - \mathbf{x}_t\|_{G_t}^2\right) - (\mathbf{x}_{t+1} - \mathbf{x})^\top A^\top (\beta(A\mathbf{x}_{t+1} - \mathbf{y}_t) - \lambda_t)$$

$$- (\mathbf{x}_{t+1} - \mathbf{x})^\top A^\top \lambda_{t+1} + \mathbf{g}_t^\top (\mathbf{x}_t - \mathbf{x}_{t+1}) + \Delta_t$$

$$= \frac{1}{2\eta}\left(\|\mathbf{x}_t - \mathbf{x}\|_{G_t}^2 - \|\mathbf{x}_{t+1} - \mathbf{x}\|_{G_t}^2 - \|\mathbf{x}_{t+1} - \mathbf{x}_t\|_{G_t}^2\right) - (\mathbf{x}_{t+1} - \mathbf{x})^\top A^\top (\beta(A\mathbf{x}_{t+1} - \mathbf{y}_t) - \lambda_t)$$

$$- (\mathbf{x}_{t+1} - \mathbf{x})^\top A^\top (\lambda_t - \beta(A\mathbf{x}_{t+1} - \mathbf{y}_{t+1})) + \mathbf{g}_t^\top (\mathbf{x}_t - \mathbf{x}_{t+1}) + \Delta_t$$

$$= \frac{1}{2\eta}\left(\|\mathbf{x}_t - \mathbf{x}\|_{G_t}^2 - \|\mathbf{x}_{t+1} - \mathbf{x}\|_{G_t}^2 - \|\mathbf{x}_{t+1} - \mathbf{x}_t\|_{G_t}^2\right) - (\mathbf{x}_{t+1} - \mathbf{x})^\top A^\top \beta(\mathbf{y}_{t+1} - \mathbf{y}_t) + \frac{\|\mathbf{x}_t - \mathbf{x}_{t+1}\|_{G_t}^2}{2\eta}$$

$$+ \frac{\eta}{2}\|\mathbf{g}_t\|_{G_t^{-1}}^2 + \Delta_t$$

$$= \frac{1}{2\eta}\left((\|\mathbf{x}_t - \mathbf{x}\|_{G_t}^2 - \|\mathbf{x}_{t+1} - \mathbf{x}\|_{G_t}^2) + \frac{\eta}{2}\|\mathbf{g}_t\|_{G_t^{-1}}^2 + \Delta_t + (\mathbf{x}_{t+1} - \mathbf{x})^\top A^\top \beta(\mathbf{y}_t - \mathbf{y}_{t+1}).\right.$$

To handle the last term in the previous inequality, we observe that

$$(\mathbf{x}_{t+1} - \mathbf{x})^\top A^\top \beta(\mathbf{y}_t - \mathbf{y}_{t+1}) = \beta(A\mathbf{x}_{t+1} - A\mathbf{x})^\top (\mathbf{y}_t - \mathbf{y}_{t+1})$$

$$= \frac{\beta}{2}\left[\|A\mathbf{x} - \mathbf{y}_t\|_2^2 - \|A\mathbf{x} - \mathbf{y}_{t+1}\|_2^2 + \|A\mathbf{x}_{t+1} - \mathbf{y}_{t+1}\|_2^2 - \|A\mathbf{x}_{t+1} - \mathbf{y}_t\|_2^2\right]$$

$$\leq \frac{\beta}{2}\left[\|A\mathbf{x} - \mathbf{y}_t\|_2^2 - \|A\mathbf{x} - \mathbf{y}_{t+1}\|_2^2\right] + \frac{1}{2\beta}\|\lambda_{t+1} - \lambda_t\|_2^2.$$

Thus we have

$$f(\mathbf{x}_t) - f(\mathbf{x}) - (\mathbf{x}_{t+1} - \mathbf{x})^\top A^\top \lambda_{t+1}$$

$$\leq \frac{1}{2\eta} \left( (\|\mathbf{x}_t - \mathbf{x}\|_{G_t}^2 - \|\mathbf{x}_{t+1} - \mathbf{x}\|_{G_t}^2) + \frac{\eta}{2} \|\mathbf{g}_t\|_{G_t^{-1}}^2 + \Delta_t \right. \tag{8}$$

$$+ \frac{\beta}{2} \left[ \|A\mathbf{x} - \mathbf{y}_t\|_2^2 - \|A\mathbf{x} - \mathbf{y}_{t+1}\|_2^2 \right] + \frac{1}{2\beta} \|\lambda_{t+1} - \lambda_t\|_2^2.$$

Next by the optimality condition of $\mathbf{y}_{t+1}$, there exists $\partial \psi(\mathbf{y}_{t+1})$ such that

$$(\mathbf{y} - \mathbf{y}_{t+1})^\top (\partial \psi(\mathbf{y}_{t+1}) + \lambda_t - \beta(A\mathbf{x}_{t+1} - \mathbf{y}_{t+1})) \geq 0.$$

Hence,

$$\psi(\mathbf{y}_{t+1}) - \psi(\mathbf{y}) \leq -(\mathbf{y} - \mathbf{y}_{t+1})^\top \partial \psi(\mathbf{y}_{t+1}) \leq (\mathbf{y} - \mathbf{y}_{t+1})^\top (\lambda_t - \beta(A\mathbf{x}_{t+1} - \mathbf{y}_{t+1}))$$

$$= (\mathbf{y} - \mathbf{y}_{t+1})^\top \lambda_{t+1}. \tag{9}$$

By the updating rule of $\lambda_{t+1}$, we have

$$(\lambda_{t+1} - \lambda)^\top (A\mathbf{x}_{t+1} - \mathbf{y}_{t+1}) = \frac{1}{\beta} (\lambda_{t+1} - \lambda)^\top (\lambda_t - \lambda_{t+1})$$

$$= \frac{1}{2\beta} \left( \|\lambda - \lambda_t\|^2 - \|\lambda - \lambda_{t+1}\|^2 - \|\lambda_t - \lambda_{t+1}\|_2^2 \right). \tag{10}$$

Adding the three inequalities in (8), (9) and (10) gives

$$f(\mathbf{x}_t) - f(\mathbf{x}) + \psi(\mathbf{y}_{t+1}) - \psi(\mathbf{y}) + (\mathbf{x}_{t+1} - \mathbf{x})^\top (-A^\top \lambda_{t+1}) + (\mathbf{y}_{t+1} - \mathbf{y})^\top \lambda_{t+1}$$

$$+ (\lambda_{t+1} - \lambda)^\top (A\mathbf{x}_{t+1} - \mathbf{y}_{t+1}) \leq \frac{1}{2\eta} \left( (\|\mathbf{x}_t - \mathbf{x}\|_{G_t}^2 - \|\mathbf{x}_{t+1} - \mathbf{x}\|_{G_t}^2) + \frac{\eta}{2} \|\mathbf{g}_t\|_{G_t^{-1}}^2 + \Delta_t \right.$$

$$+ \frac{1}{2\beta} \left( \|\lambda - \lambda_t\|^2 - \|\lambda - \lambda_{t+1}\|^2 \right) + \frac{\beta}{2} \left[ \|A\mathbf{x} - \mathbf{y}_t\|_2^2 - \|A\mathbf{x} - \mathbf{y}_{t+1}\|_2^2 \right],$$

which can be written as

$$f(\mathbf{x}_t) - f(\mathbf{x}) + \psi(\mathbf{y}_{t+1}) - \psi(\mathbf{y}) + (\mathbf{u}_{t+1} - \mathbf{u})^\top \mathcal{F}(\mathbf{u}_{t+1})$$

$$\leq \frac{1}{2\eta} \left( (\|\mathbf{x}_t - \mathbf{x}\|_{G_t}^2 - \|\mathbf{x}_{t+1} - \mathbf{x}\|_{G_t}^2) + \frac{\eta}{2} \|\mathbf{g}_t\|_{G_t^{-1}}^2 + \Delta_t + \frac{1}{2\beta} \left( \|\lambda - \lambda_t\|^2 - \|\lambda - \lambda_{t+1}\|^2 \right) \right.$$

$$+ \frac{\beta}{2} \left[ \|A\mathbf{x} - \mathbf{y}_t\|_2^2 - \|A\mathbf{x} - \mathbf{y}_{t+1}\|_2^2 \right].$$

Taking the summation over $t = 1, \ldots, T$, we have

$$\sum_{t=1}^{T} (f(\mathbf{x}_t) - f(\mathbf{x})) + \sum_{t=1}^{T} (\psi(\mathbf{y}_{t+1}) - \psi(\mathbf{y})) + \sum_{t=1}^{T} (\mathbf{u}_{t+1} - \mathbf{u})^\top \mathcal{F}(\mathbf{u}_{t+1})$$

$$\leq \frac{1}{2\eta} \sum_{t=1}^{T} \left( \|\mathbf{x}_t - \mathbf{x}\|_{G_t}^2 - \|\mathbf{x}_{t+1} - \mathbf{x}\|_{G_t}^2 \right) + \sum_{t=1}^{T} \frac{\eta \|\mathbf{g}_t\|_{G_t^{-1}}^2}{2} + \frac{1}{2\beta} \|\lambda - \lambda_1\|_2^2 + \sum_{t=1}^{T} \Delta_t$$

$$+ \frac{\beta}{2} \|A\mathbf{x} - \mathbf{y}_1\|_2^2.$$

By the convexity of $f(\mathbf{x})$ and $\psi(y)$ and the monotonicity of $\mathcal{F}(\cdot)$, we have

$$f(\widehat{\mathbf{x}}_T) - f(\mathbf{x}) + \psi(\widehat{\mathbf{y}}_T) - \psi(\mathbf{y}) + (\widehat{\mathbf{u}}_T - \mathbf{u})^\top \mathcal{F}(\widehat{\mathbf{u}}_T)$$

$$\leq \frac{1}{T} \left( \sum_{t=1}^{T} (f(\mathbf{x}_t) - f(\mathbf{x})) + \sum_{t=1}^{T} (\psi(\mathbf{y}_{t+1}) - \psi(\mathbf{y})) + \sum_{t=1}^{T} (\mathbf{u}_{t+1} - \mathbf{u})^\top \mathcal{F}(\mathbf{u}_{t+1}) \right)$$

$$\leq \frac{1}{2\eta T} \sum_{t=1}^{T} \left( \|\mathbf{x}_t - \mathbf{x}\|_{G_t}^2 - \|\mathbf{x}_{t+1} - \mathbf{x}\|_{G_t}^2 \right) + \frac{\eta}{2T} \sum_{t=1}^{T} \|\mathbf{g}_t\|_{G_t^{-1}}^2 + \frac{1}{2\beta T} \|\lambda - \lambda_1\|_2^2$$

$$+ \frac{\beta}{2T} \|A\mathbf{x} - \mathbf{y}_1\|_2^2 + \frac{1}{T} \sum_{t=1}^{T} \Delta_t. \tag{11}$$

Due to the fact that

$$(\widehat{\mathbf{u}}_T - \mathbf{u})^\top \mathcal{F}(\widehat{\mathbf{u}}_T) = (\bar{\mathbf{x}}_T - \mathbf{x})^\top (-A^\top \widehat{\lambda}_T) + (\widehat{\mathbf{y}}_T - \mathbf{y})^\top \widehat{\lambda}_T + (\widehat{\lambda}_T - \lambda)(A\bar{\mathbf{x}}_T - \widehat{\mathbf{y}}_T)$$
$$= \widehat{\lambda}_T^\top (A\mathbf{x} - \mathbf{y}) - \lambda^\top (A\bar{\mathbf{x}}_T - \widehat{\mathbf{y}}_T) = -\lambda^\top (A\bar{\mathbf{x}}_T - \widehat{\mathbf{y}}_T),$$

we have

$$\max_{\|\lambda\|_2 \le \rho} (\widehat{\mathbf{u}}_T - \mathbf{u})^\top \mathcal{F}(\widehat{\mathbf{u}}_T) = \rho \|A\bar{\mathbf{x}}_T - \widehat{\mathbf{y}}_T\|_2,$$

which, by Assumption 1.c, implies

$$\psi(A\bar{\mathbf{x}}_T) - \psi(\widehat{\mathbf{y}}_T) \le \rho \|A\bar{\mathbf{x}}_T - \widehat{\mathbf{y}}_T\|_2 \le \max_{\|\lambda\|_2 \le \rho} (\widehat{\mathbf{u}}_T - \mathbf{u})^\top \mathcal{F}(\widehat{\mathbf{u}}_T).$$

According to this inequality and the fact that $\lambda_1 = 0$, if we fix $\mathbf{x}$ and $\mathbf{y} = A\mathbf{x}$ but change $\lambda$ to maximize both sides of (11) over $\|\lambda\|_2 \le \rho$, we obtain

$$f(\widehat{\mathbf{x}}_T) - f(\mathbf{x}) + \psi(A\bar{\mathbf{x}}_T) - \psi(\mathbf{y})$$
$$\le \frac{1}{2\eta T} \sum_{t=1}^{T} \left( \|\mathbf{x}_t - \mathbf{x}\|_{G_t}^2 - \|\mathbf{x}_{t+1} - \mathbf{x}\|_{G_t}^2 \right) + \frac{\eta}{2T} \sum_{t=1}^{T} \|\mathbf{g}_t\|_{G_t^{-1}}^2 + \frac{\rho^2}{2\beta T}$$
$$+ \frac{\beta}{2T} \|A\mathbf{x} - \mathbf{y}_1\|_2^2 + \frac{1}{T} \sum_{t=1}^{T} \Delta_t.$$

Adding $\psi(A\widehat{\mathbf{x}}_T) - \psi(A\bar{\mathbf{x}}_T)$ to both sides of this inequality leads to

$$f(\widehat{\mathbf{x}}_T) - f(\mathbf{x}) + \psi(A\widehat{\mathbf{x}}_T) - \psi(\mathbf{y})$$
$$\le \frac{1}{2\eta T} \sum_{t=1}^{T} \left( \|\mathbf{x}_t - \mathbf{x}\|_{G_t}^2 - \|\mathbf{x}_{t+1} - \mathbf{x}\|_{G_t}^2 \right) + \frac{\eta}{2T} \sum_{t=1}^{T} \|\mathbf{g}_t\|_{G_t^{-1}}^2 + \frac{\rho^2}{2\beta T}$$
$$+ \frac{\beta}{2T} \|A\mathbf{x} - \mathbf{y}_1\|_2^2 + \frac{1}{T} \sum_{t=1}^{T} \Delta_t + \psi(A\widehat{\mathbf{x}}_T) - \psi(A\bar{\mathbf{x}}_T)$$
$$\le \frac{1}{2\eta T} \sum_{t=1}^{T} \left( \|\mathbf{x}_t - \mathbf{x}\|_{G_t}^2 - \|\mathbf{x}_{t+1} - \mathbf{x}\|_{G_t}^2 \right) + \frac{\eta}{2T} \sum_{t=1}^{T} \|\mathbf{g}_t\|_{G_t^{-1}}^2 + \frac{\rho^2}{2\beta T}$$
$$+ \frac{\beta}{2T} \|A\mathbf{x} - \mathbf{y}_1\|_2^2 + \frac{1}{T} \sum_{t=1}^{T} \Delta_t + \rho \|A(\bar{\mathbf{x}}_T - \widehat{\mathbf{x}}_T)\|_2$$
$$\le \frac{1}{2\eta T} \sum_{t=1}^{T} \left( \|\mathbf{x}_t - \mathbf{x}\|_{G_t}^2 - \|\mathbf{x}_{t+1} - \mathbf{x}\|_{G_t}^2 \right) + \frac{\eta}{2T} \sum_{t=1}^{T} \|\mathbf{g}_t\|_{G_t^{-1}}^2 + \frac{\rho^2}{2\beta T}$$
$$+ \frac{\beta}{2T} \|A\mathbf{x} - \mathbf{y}_1\|_2^2 + \frac{1}{T} \sum_{t=1}^{T} \Delta_t + \frac{\rho \|A(\mathbf{x}_1 - \mathbf{x}_{T+1})\|_2}{T},$$

where the second inequality is a result of Assumption 1.c and the third inequality is by the definition of $\bar{\mathbf{x}}_T$ and $\widehat{\mathbf{x}}_T$. □

## 6 Practical Variants of Locally Adaptive ADMM

In this section, we present variants of locally adaptive ADMM algorithms that can be implemented with unknown constant $c$ and unknown exponent parameter $\theta$. Following the idea of [2], we propose using another level of restarting on our ADMM. In particular, we apply our ADMM method in epochs where we start the first epoch with a relatively large number of iterations $t_1$ and then after each epoch we increase it gradually. We present the detailed steps in Algorithms 5 and 6, and the convergence results in Theorems 6 and 8 for unknown $c$ but known $\theta \in (0,1)$ and in Theorems 9 and 10 for unknown $c$ and $\theta$.

## 6.1 Locally Adaptive ADMM for unknown $c$

When the constant $c$ is unknown but $\theta \in (0,1)$ is known, we present formal guarantee of Algorithm 5 in the following theorem.

**Theorem 6** (RLA-ADMM with unknown $c$). *Suppose Assumption 1 holds. Let $\epsilon \leq \epsilon_0/4$ and $K = \lceil \log_2(\frac{\epsilon_0}{\epsilon}) \rceil$ in Algorithm 5. Suppose $\beta_1^{(1)}$ is sufficiently small so that there exists $\hat{\epsilon}_1 \in [\epsilon, \epsilon_0/2]$, with which $F(\cdot)$ satisfies a local error bound condition on $S_{\hat{\epsilon}_1}$ with $\theta \in (0,1)$ and the constant $c$, and $\beta_1^{(1)} = \frac{\sqrt{2}\rho\hat{\epsilon}_1^{1-\theta}}{c\|A\|_2\epsilon_0}$. Let $t_1 = \left\lceil \frac{2\rho^2}{\beta_1^{(1)}\epsilon_0} \right\rceil$ and $S = \lceil \log_2(\hat{\epsilon}_1/\epsilon) \rceil + 1$. Then, with a total number of $S$ calls of RLA-ADMM in Algorithm 5, we find a solution $\mathbf{x}^{(S)}$ such that $F(\mathbf{x}^{(S)}) - F_* \leq 2\epsilon$. The total number of iterations of RLA-ADMM for obtaining $2\epsilon$-optimal solution is upper bounded by $T_S = \widetilde{O}(1/\epsilon^{1-\theta})$.*

To prove the above theorem, we need the following theorem.

**Theorem 7.** *Suppose Assumption 1 holds and $F(\mathbf{x})$ obeys a local error bound condition on the $\epsilon$-sublevel. Let $K = \lceil \log_2(\epsilon_0/\epsilon) \rceil$, $\beta_1 = \frac{\sqrt{2}\rho\epsilon^{1-\theta}}{c\|A\|_2\epsilon_0}$ and $t = \left\lceil \frac{2\rho^2}{\beta_1\epsilon_0} \right\rceil$ in Algorithm 2, we have*

$$F(\mathbf{x}_K) - F_* \leq 2\epsilon.$$

*Proof.* Here we only prove for the case of $G = \gamma I - \beta A^\top A$ with $\gamma = \beta\|A\|_2^2$. Following the same analysis, we can easily prove the same result for using $G = 0$. Let $\mathbf{x}_{k-1,\epsilon}^\dagger$ denote the closest point to $\mathbf{x}_{k-1}$ in $S_\epsilon$. Define $\epsilon_k = \frac{\epsilon_0}{2^k}$. Then $\beta_k = \frac{\rho\epsilon^{1-\theta}}{\sqrt{2}c\|A\|_2\epsilon_k}$ and $\gamma_k = \beta_k\|A\|_2^2 = \frac{\rho\epsilon^{1-\theta}\|A\|_2}{\sqrt{2}c\epsilon_k}$. We prove this by induction. Assume $F(\mathbf{w}_{k-1}) - F_* \leq \epsilon_{k-1} + \epsilon$, which trivally holds for $k=1$ due to Assumption 1.a. We apply Corollary 1 to the $k$-th stage of LA-ADMM. For any $\mathbf{x} \in \Omega$, we have

$$F(\mathbf{x}_k) - F(\mathbf{x}) \leq \frac{\gamma_k\|\mathbf{x} - \mathbf{x}_{k-1}\|_2^2}{2t} + \frac{\beta_k\|A\|_2^2\|\mathbf{x} - \mathbf{x}_{k-1}\|_2^2}{2t} + \frac{\rho^2}{2\beta_k t}.$$

Let $\mathbf{x} = \mathbf{x}_{k-1,\epsilon}^\dagger$ so that we have

$$F(\mathbf{x}_k) - F(\mathbf{x}_{k-1,\epsilon}^\dagger) \leq \frac{\gamma_k\|\mathbf{x}_{k-1,\epsilon}^\dagger - \mathbf{x}_{k-1}\|_2^2}{2t} + \frac{\beta_k\|A\|_2^2\|\mathbf{x}_{k-1,\epsilon}^\dagger - \mathbf{x}_{k-1}\|_2^2}{2t} + \frac{\rho^2}{2\beta_k t}. \tag{12}$$

We consider two scenarios of $\mathbf{x}_{k-1}$. First, suppose $\mathbf{x}_{k-1} \in S_\epsilon$ so that $\mathbf{x}_{k-1} = \mathbf{x}_{k-1,\epsilon}^\dagger$. Then

$$F(\mathbf{x}_k) - F(\mathbf{x}_{k-1,\epsilon}^\dagger) \leq \frac{\rho^2}{2\beta_k t} \leq \frac{\rho^2\sqrt{2}c\|A\|_2\epsilon_k}{2\rho\epsilon^{1-\theta}}\frac{\epsilon^{1-\theta}}{\sqrt{2}c\rho\|A\|_2} = \frac{\epsilon_k}{2},$$

which implies

$$F(\mathbf{x}_k) - F_* \leq F(\mathbf{x}_{k-1,\epsilon}^\dagger) - F_* + \epsilon_k \leq \epsilon + \epsilon_k.$$

Secondly, suppose $\mathbf{x}_{k-1} \notin S_\epsilon$ so that $F(\mathbf{x}_{k-1,\epsilon}^\dagger) = F_* + \epsilon$. By Lemma 1 and the local error bound condition of $F$, we have

$$\|\mathbf{x}_{k-1} - \mathbf{x}_{k-1,\epsilon}^\dagger\|_2 \leq \frac{dist(\mathbf{x}_{k-1,\epsilon}^\dagger, \Omega_*)}{\epsilon}(F(\mathbf{x}_{k-1}) - F(\mathbf{x}_{k-1,\epsilon}^\dagger)) \leq \frac{c\epsilon^\theta}{\epsilon}\epsilon_{k-1} = \frac{c\epsilon_{k-1}}{\epsilon^{1-\theta}}.$$

where we use the assumption that $F(\mathbf{x}_{k-1}) - F_* \leq \epsilon + \epsilon_{k-1}$ and the fact $F(\mathbf{x}_{k-1,\epsilon}^\dagger) = F_* + \epsilon$. Plugging the above bound into (12) we have

$$F(\mathbf{x}_k) - F(\mathbf{x}_{k-1,\epsilon}^\dagger) \leq \frac{\beta_k\|A\|_2^2 c^2\epsilon_{k-1}^2}{2t\epsilon^{2(1-\theta)}} + \frac{\beta_k\|A\|_2^2 c^2\epsilon_{k-1}^2}{2t\epsilon^{2(1-\theta)}} + \frac{\rho^2}{2\beta_k t}$$

$$= \frac{\rho\epsilon^{1-\theta}\|A\|_2^2 c^2\epsilon_{k-1}^2}{\sqrt{2}c\|A\|_2\epsilon_k t\epsilon^{2(1-\theta)}} + \frac{\sqrt{2}c\rho^2\epsilon_k\|A\|_2}{2\rho\epsilon^{1-\theta}t}.$$

Since $t \geq \frac{\sqrt{2}c\rho\|A\|_2}{\epsilon^{1-\theta}}$, we have

$$F(\mathbf{x}_k) - F(\mathbf{x}_{k-1,\epsilon}^\dagger) \leq \frac{\epsilon_k}{2} + \frac{\epsilon_k}{2} = \epsilon_k,$$

---

**Algorithm 5** LA-ADMM with Restarting (RLA-ADMM)

---

1: **Input**: the number of iterations $t_1$ per epoch and the penalty parameter $\beta_1^{(1)}$ in the first stage.
2: **Initialization**: $\mathbf{x}^{(0)}$
3: **for** $s = 1, 2, \ldots,$ **do**
4:     Let $\mathbf{x}^{(s)} =$LA-ADMM$(\mathbf{x}^{(s-1)}, \beta_1^{(s)}, K, t_s)$
5:     Let $t_{s+1} = t_s 2^{1-\theta}$, $\beta_1^{(s+1)} = \beta_1^{(s)}/2^{1-\theta}$
6: **end for**
7: **Output**: $\mathbf{x}^{(S)}$

---

---

**Algorithm 6** LA-SADMM with Restarting (RLA-SADMM)

---

1: **Input**: the number of iterations $t_1$ per epoch and the radius $D_1^{(1)}$ in the first stage.
2: **Initialization**: $\mathbf{x}^{(0)}$, $\eta_1 = \frac{\epsilon_0}{6R^2}$, $\beta_1 = \frac{6R^2}{\|A\|_2^2 \epsilon_0}$
3: **for** $s = 1, 2, \ldots,$ **do**
4:     Let $\mathbf{x}^{(s)} =$LA-SADMM$(\mathbf{x}^{(s-1)}, D_1^{(s)}, K, t_s)$
5:     Let $t_{s+1} = t_s 2^{2(1-\theta)}$, $D_1^{(s+1)} = D_1^{(s)} 2^{1-\theta}$
6: **end for**
7: **Output**: $\mathbf{x}^{(S)}$

---

which implies

$$F(\mathbf{x}_k) - F_* \leq \epsilon_k + \epsilon.$$

We can finish the proof by the induction up to $k = K = \lceil \log_2(\epsilon_0/\epsilon) \rceil$, which yields $F(\mathbf{x}_K) - F_* \leq 2\epsilon$. $\qquad \square$

*Proof of Theorem 6.* Following the proof of Theorem 7, we can show that

$$F(\mathbf{x}^{(1)}) - F_* \leq 2\hat{\epsilon}_1 \leq \epsilon_0. \qquad (13)$$

with $K = \lceil \log_2(\frac{\epsilon_0}{\epsilon}) \rceil \geq \lceil \log_2(\frac{\epsilon_0}{\hat{\epsilon}_1}) \rceil$, $\beta_1^{(1)} = \frac{\sqrt{2}\rho\hat{\epsilon}_1^{1-\theta}}{c\|A\|_2 \epsilon_0}$, and $t_1 = \left\lceil \frac{2\rho^2}{\beta_1^{(1)}\epsilon_0} \right\rceil$. By running LA-ADMM starting from $\mathbf{x}^{(1)}$ which satisfies (13) with $K = \lceil \log_2(\frac{\epsilon_0}{\epsilon}) \rceil \geq \lceil \log_2(\frac{\epsilon_0}{\hat{\epsilon}_1/2}) \rceil$, $\beta_1^{(2)} = \beta_1^{(1)}/2^{1-\theta} = \frac{\sqrt{2}\rho(\hat{\epsilon}_1/2)^{1-\theta}}{c\|A\|_2 \epsilon_0}$ and $t_2 = t_1 2^{1-\theta} = \left\lceil \frac{2\rho^2}{\beta_1^{(2)}\epsilon_0} \right\rceil$, Theorem 7 ensures that

$$F(\mathbf{x}^{(2)}) - F_* \leq \hat{\epsilon}_1 \leq \epsilon_0.$$

Applying this argument recursively, we can show

$$F(\mathbf{x}^{(s)}) - F_* \leq 2\hat{\epsilon}_1/2^{s-1} \leq \epsilon_0, \text{ for } s = 1, 2, \ldots.$$

With $S = \lceil \log_2(\hat{\epsilon}_1/\epsilon) \rceil + 1$, we prove that

$$F(\mathbf{x}^{(S)}) - F_* \leq 2\hat{\epsilon}_1/2^{S-1} \leq 2\epsilon.$$

The total number of iterations for the $S$ calls of LA-ADMM is bounded by

$$T_S = K \sum_{s=1}^{S} T_s = K \sum_{s=1}^{S} t_1 2^{(s-1)(1-\theta)} = K t_1 2^{(S-1)(1-\theta)} \sum_{s=1}^{S} \left(1/2^{(1-\theta)}\right)^{S-s}$$

$$\leq K t_1 2^{(S-1)(1-\theta)} \frac{1}{1 - 1/2^{(1-\theta)}} \leq O\left(K t_1 \left(\frac{\hat{\epsilon}_1}{\epsilon}\right)^{(1-\theta)}\right) \leq \widetilde{O}(1/\epsilon^{(1-\theta)}).$$

$\qquad \square$

When the constant $c$ is unknown but $\theta \in (0, 1)$ is known, the formal guarantee of Algorithm 6 is presented in the following theorem.

**Theorem 8** (RFA-SADMM with unknown $c$). *Suppose Assumptions 1 and 2 hold. Let $\epsilon \leq \epsilon_0/4$ and $K = \lceil \log_2(\frac{\epsilon_0}{\epsilon}) \rceil$ in Algorithm 6. Suppose $D_1^{(1)}$ is sufficiently large so that there exists $\hat{\epsilon}_1 \in [\epsilon, \epsilon_0/2]$, with which $F(\cdot)$ satisfies a local error bound condition on $\mathcal{S}_{\hat{\epsilon}_1}$ with $\theta \in (0,1)$ and the constant $c$, and $D_1^{(1)} = \frac{c\epsilon_0}{\hat{\epsilon}_1^{1-\theta}}$. Let $t_1 = \max\left\{ \frac{6912R^2 \log(1/\tilde{\delta})(D_1^{(1)})^2}{\epsilon_0^2}, \frac{12\rho\|A\|_2 D_1^{(1)}}{\epsilon_0}, \frac{\rho^2\|A\|_2^2}{R^2} \right\}$, $S = \lceil \log_2(\hat{\epsilon}_1/\epsilon) \rceil + 1$ and $\hat{\delta} = \frac{\delta}{KS}$. Then, with a total number of $S$ calls of LA-SADMM in Algorithm 6, we find a solution $\mathbf{x}^{(S)}$ such that $F(\mathbf{x}^{(S)}) - F_* \leq 2\epsilon$. The total number of iterations of RLA-SADMM for obtaining $2\epsilon$-optimal solution is upper bounded by $T_S = \widetilde{O}(\log(1/\delta)/\epsilon^{2(1-\theta)})$.*

*Proof.* With $K = \lceil \log_2(\frac{\epsilon_0}{\epsilon}) \rceil \geq \lceil \log_2(\frac{\epsilon_0}{\hat{\epsilon}_1}) \rceil$ and

$$t_1 = \max\left\{ \frac{6912R^2 \log(1/\tilde{\delta})(D_1^{(1)})^2}{\epsilon_0^2}, \frac{12\rho\|A\|_2 D_1^{(1)}}{\epsilon_0}, \frac{\rho^2\|A\|_2^2}{R^2} \right\}$$

and $D_1^{(1)} = \frac{c\epsilon_0}{\hat{\epsilon}_1^{1-\theta}}$, by Theorem 4, we can show that

$$F(\mathbf{x}^{(1)}) - F_* \leq 2\hat{\epsilon}_1 \leq \epsilon_0 \tag{14}$$

with a probability of at least $1 - \frac{\delta}{S}$. By running RLA-SADMM starting from $\mathbf{x}^{(1)}$ which satisfies (14) with $K = \lceil \log_2(\frac{\epsilon_0}{\epsilon}) \rceil \geq \lceil \log_2(\frac{2\hat{\epsilon}_1}{\hat{\epsilon}_1/2}) \rceil$ and

$$t_2 = t_1 2^{2(1-\theta)} \geq \max\left\{ \frac{6912R^2 \log(1/\tilde{\delta})(D_1^{(2)})^2}{\epsilon_0^2}, \frac{12\rho\|A\|_2 D_1^{(2)}}{\epsilon_0}, \frac{\rho^2\|A\|_2^2}{R^2} \right\}$$

with $D_1^{(2)} = \frac{c\epsilon_0}{(\hat{\epsilon}_1/2)^{1-\theta}} \geq \frac{c2\hat{\epsilon}_1}{(\hat{\epsilon}_1/2)^{1-\theta}}$, Theorem 4 ensures that

$$F(\mathbf{x}^{(2)}) - F_* \leq \hat{\epsilon}_1 \leq \epsilon_0$$

with a probality of at least $(1 - \delta/S)^2$. Applying this argument repeatedly, we have

$$F(\mathbf{x}^{(s)}) - F_* \leq 2\hat{\epsilon}_1/2^{s-1} \leq \epsilon_0, \text{ for } s = 1, 2, \ldots, S$$

with a probality of at least $(1 - \delta/S)^s$. With $S = \lceil \log_2(\hat{\epsilon}_1/\epsilon) \rceil + 1$, we can prove that, with a probality of at least $(1 - \delta/S)^S \geq 1 - \delta$,

$$F(\mathbf{x}^{(S)}) - F_* \leq 2\hat{\epsilon}_1/2^{S-1} \leq 2\epsilon.$$

The total number of iterations for the $S$ calls of RLA-SADMM is bounded by

$$T_S = K \sum_{s=1}^{S} T_s = K \sum_{s=1}^{S} t_1 2^{2(s-1)(1-\theta)} = Kt_1 2^{2(S-1)(1-\theta)} \sum_{s=1}^{S} \left( 1/2^{2(1-\theta)} \right)^{S-s}$$

$$\leq Kt_1 2^{2(S-1)(1-\theta)} \frac{1}{1 - 1/2^{2(1-\theta)}} \leq O\left( Kt_1 \left( \frac{\hat{\epsilon}_1}{\epsilon} \right)^{2(1-\theta)} \right) \leq \widetilde{O}(\log(1/\delta)/\epsilon^{2(1-\theta)}).$$

$\square$

## 6.2 Locally Adaptive ADMM for unknown $\theta$

In this subsection, we show that the iteration complexity of the proposed algorithms can be no worse than standard ADMM algorithms even the value of $\theta$ is unknown. If the exponent parameter $\theta$ is unkown, we observe that $F(\cdot)$ satisfies a local error bound condition on $\mathcal{S}_\epsilon$ with $\theta = 0$ and $c = B_{\epsilon'}$ with $\epsilon' > \epsilon$ for any $\epsilon > 0$. Then, the following theorem is derived based on this observation.

**Theorem 9** (RLA-ADMM with unknown $\theta$). *Let $\theta = 0$, $\epsilon \leq \epsilon_0/4$ and $K = \lceil \log_2(\frac{\epsilon_0}{\epsilon}) \rceil$ in Algorithm 5. Assume $\beta_1^{(1)}$ is sufficiently small such that there exists $\hat{\epsilon}_1 \in [\epsilon, \epsilon_0/2]$ rendering $\beta_1^{(1)} = \frac{\sqrt{2}\rho\hat{\epsilon}_1}{B_{\hat{\epsilon}_1}\|A\|_2 \epsilon_0}$. Let $t_1 = \left\lceil \frac{2\rho^2}{\beta_1^{(1)} \epsilon_0} \right\rceil$ and $S = \lceil \log_2(\hat{\epsilon}_1/\epsilon) \rceil + 1$. Then, with a total number of $S$ calls of LA-ADMM in Algorithm 5, we find a solution $\mathbf{x}^{(S)}$ such that $F(\mathbf{x}^{(S)}) - F_* \leq 2\epsilon$. The total number of iterations of RLA-ADMM for obtaining $2\epsilon$-optimal solution is upper bounded by $T_S = \widetilde{O}\left( \frac{\rho\|A\|_2 B_{\hat{\epsilon}_1}}{\epsilon} \right)$.*

**Remark:** Compared to the standard ADMM (see remark below Corollary 1), we can see that RLA-ADMM converges no slower than standard ADMM as long as $\beta_1^{(1)}$ is sufficiently small.

*Proof.* This proof is similar to the proof of Theorem 6 except that $\theta = 0$ and $c = B_{\hat{\epsilon}_1}$. Given that $K = \lceil \log_2(\frac{\epsilon_0}{\epsilon}) \rceil \geq \lceil \log_2(\frac{\epsilon_0}{\hat{\epsilon}_1}) \rceil$, $t_1 = \left\lceil \frac{\sqrt{2} B_{\hat{\epsilon}_1} \rho \|A\|_2}{\hat{\epsilon}_1} \right\rceil$ and $\beta_1^{(1)} = \frac{\sqrt{2} \rho \hat{\epsilon}_1}{B_{\hat{\epsilon}_1} \|A\|_2 \epsilon_0}$, by Theorem 7, we have

$$F(\mathbf{x}^{(1)}) - F_* \leq 2\hat{\epsilon}_1 \leq \epsilon_0. \tag{15}$$

By running LA-ADMM starting from $\mathbf{x}^{(1)}$ which satisfies (15) with $K = \lceil \log_2(\frac{\epsilon_0}{\epsilon}) \rceil \geq \lceil \log_2(\frac{\epsilon_0}{\hat{\epsilon}_1/2}) \rceil$, $t_2 = t_1 2 = \left\lceil \frac{\sqrt{2} B_{\hat{\epsilon}_1} \rho \|A\|_2}{(\hat{\epsilon}_1/2)} \right\rceil$, and $\beta_1^{(2)} = \beta_1^{(1)}/2 = \frac{\sqrt{2} \rho (\hat{\epsilon}_1/2)}{B_{\hat{\epsilon}_1} \|A\|_2 \epsilon_0}$, Theorem 7 ensures that

$$F(\mathbf{x}^{(2)}) - F_* \leq \hat{\epsilon}_1 \leq \epsilon_0.$$

Applying this argument repeatedly, with $S = \lceil \log_2(\hat{\epsilon}_1/\epsilon) \rceil + 1$ we can prove that

$$F(\mathbf{x}^{(S)}) - F_* \leq \hat{2}\epsilon_1/2^{S-1} \leq 2\epsilon.$$

The total number of iterations for the $S$ calls of LA-ADMM is bounded by

$$T_S = K \sum_{s=1}^{S} T_s = K \sum_{s=1}^{S} t_1 2^{(s-1)} = K t_1 2^{(S-1)} \sum_{s=1}^{S} (1/2)^{S-s}$$

$$\leq K t_1 2^{(S-1)} \frac{1}{1 - 1/2} \leq O\left( K t_1 \frac{\hat{\epsilon}_1}{\epsilon} \right) = O\left( \frac{B_{\hat{\epsilon}_1} \|A\|_2}{\epsilon} \lceil \log_2(\frac{\epsilon_0}{\epsilon}) \rceil \right).$$

$\square$

**Theorem 10** (RLA-SADMM with unknown $\theta$). *Let $\theta = 0$, $\epsilon \leq \epsilon_0/4$ and $K = \lceil \log_2(\frac{\epsilon_0}{\epsilon}) \rceil$ in Algorithm 6. Assume $D_1^{(1)}$ is sufficiently large such that there exists $\hat{\epsilon}_1 \in [\epsilon, \epsilon_0/2]$ rendering $D_1^{(1)} = \frac{B_{\hat{\epsilon}_1} \epsilon_0}{\hat{\epsilon}_1}$. Let $t_1 = \max\left\{ \frac{6912 R^2 \log(1/\tilde{\delta})(D_1^{(1)})^2}{\epsilon_0^2}, \frac{12\rho\|A\|_2 D_1^{(1)}}{\epsilon_0}, \frac{\rho^2 \|A\|_2^2}{R^2} \right\}$, $S = \lceil \log_2(\hat{\epsilon}_1/\epsilon) \rceil + 1$ and $\hat{\delta} = \frac{\delta}{KS}$. Then, with a total number of $S$ calls of LA-SADMM in Algorithm 6, we find a solution $\mathbf{x}^{(S)}$ such that $F(\mathbf{x}^{(S)}) - F_* \leq 2\epsilon$. The total number of iterations of RLA-SADMM for obtaining $2\epsilon$-optimal solution is upper bounded by $T_S = \widetilde{O}(\log(1/\delta)/\epsilon^2)$.*

*Proof.* This proof is quite similar to that of Theorem 8 except for setting $\theta = 0$ and $c = B_{\hat{\epsilon}_1}$. Given $K = \lceil \log_2(\frac{\epsilon_0}{\epsilon}) \rceil \geq \lceil \log_2(\frac{\epsilon_0}{\hat{\epsilon}_1}) \rceil$ and

$$t_1 = \max\left\{ \frac{6912 R^2 \log(1/\tilde{\delta})(D_1^{(1)})^2}{\epsilon_0^2}, \frac{12\rho\|A\|_2 D_1^{(1)}}{\epsilon_0}, \frac{\rho^2 \|A\|_2^2}{R^2} \right\},$$

where $D_1^{(1)} = \frac{B_{\hat{\epsilon}_1} \epsilon_0}{\hat{\epsilon}_1}$, following the proof of Theorem 4, we can show that with a probability $1 - \frac{\delta}{S}$,

$$F(\mathbf{w}^{(1)}) - F_* \leq 2\hat{\epsilon}_1 \leq \epsilon_0. \tag{16}$$

By running RLA-SADMM starting from $\mathbf{x}^{(1)}$ which satisfies (16) with $K = \lceil \log_2(\frac{\epsilon_0}{\epsilon}) \rceil \geq \lceil \log_2(\frac{2\hat{\epsilon}_1}{\hat{\epsilon}_1/2}) \rceil$, $t_2 = t_1 2^2 \geq \max\left\{ \frac{6912 R^2 \log(1/\tilde{\delta})(D_1^{(2)})^2}{\epsilon_0^2}, \frac{12\rho\|A\|_2 D_1^{(2)}}{\epsilon_0}, \frac{\rho^2 \|A\|_2^2}{R^2} \right\}$ and $D_1^{(2)} = \frac{B_{\hat{\epsilon}_1} \epsilon_0}{(\hat{\epsilon}_1/2)} \geq \frac{B_{\hat{\epsilon}_1} 2\hat{\epsilon}_1}{(\hat{\epsilon}_1/2)}$, Theorem 4 ensures that

$$F(\mathbf{x}^{(2)}) - F_* \leq \hat{\epsilon}_1.$$

with a probability of at least $(1 - \delta/S)^2$. Applying this argument repeatedly, we can prove that, with a probability of at least $(1 - \delta/S)^s$,

$$F(\mathbf{x}^{(S)}) - F_* \leq 2\hat{\epsilon}_1/2^{s-1} \leq 2\epsilon, \text{ for } s = 1, 2, \ldots.$$

Figure 2: Comparison of different algorithms for solving different tasks. RR + LR represents robust regression with a low rank regularizer. LRR represents low-rank representation.

Let $S = \lceil \log_2(\hat{\epsilon}_1/\epsilon) \rceil + 1$, we have

$$F(\mathbf{x}^{(S)}) - F_* \leq 2\hat{\epsilon}_1/2^{S-1} \leq 2\epsilon$$

holds with a probability of at least $(1 - \delta/S)^S \geq 1 - \delta$. The total number of iterations for the $S$ calls of RLA-SADMM is bounded by

$$T_S = K \sum_{s=1}^{S} T_s = K \sum_{s=1}^{S} t_1 2^{2(s-1)} = K t_1 2^{2(S-1)} \sum_{s=1}^{S} \left(1/2^2\right)^{S-s}$$

$$\leq \frac{K t_1 2^{2(S-1)}}{1 - 1/2^2} \leq O\left(K t_1 \left(\frac{\hat{\epsilon}_1}{\epsilon}\right)^2\right) \leq \widetilde{O}(\log(1/\delta)/\epsilon^2).$$

$\square$

# 7 Additional Experiments

To examine the convergence behavior of different algorithms in terms of running time (cpu time), we provide the running time results in Figure 2. The results indicate that our methods are much faster than their corresponding baselines, which is similar to the results in Figure 1.