[Reviews · NeurIPS 2017]

Reviewer 1



The paper considers to accelerate ADMM using adaptive penalty parameter. The paper developed a new adaptive scheme of penalty parameters for ADMM and established theoretical guarantees for both deterministic and stochastic ADMMs. Overall, the paper is interesting in the use of local error bound but the contribution is not significant enough. 1. The new adaptive penalization scheme turns out to be loop warm start algorithms. The algorithm could be practically slow, though faster convergence is claimed. 2. Increasing penalty parameter after each outer-loop iterate could greatly slow down the algorithm when K is large.

Reviewer 2



Summary: This paper shows that O(1/eps) iteration complexity of ADMM can be improved to O(1/eps^(1-theta)) where theta is a parameter that characterizes how sharply the objective function increases with respect to increasing distance to the optimal solution. This improvement is shown under a locally adaptive version of the ADMM where the penalty parameter is increased after every $t$ steps of ADMM. The method is extended to stochastic ADMM whose O(1/eps^2) iteration complexity is shown to similarly improve. The results are backed by experiments on generalized Lasso problems. Overall, the paper is well written and makes an important contribution towards improving the analysis of ADMM under adaptive penalty parameters. On a practical note, I would have liked to see some comparisons against other "adaptive-rho" heuristics used the literature (see Boyd's monograph). As background, it may be valuable to some readers to see how the local error bound relates to the KL property, and some examples of "theta" for problems of interest in machine learning. There are lots of grammatical errors, typos and odd phrases in the the abstract: "tremendous interests" --> "tremendous interest", "variants are"-->"variants is", "penalty scheme of lies at it is...", "iterate message".. The LADMM-AP behavior in plot 1(e) is somewhat strange. Any explanations? If factor of 2 in LA-ADMM optimal in any sense? In practice ADMM stopping criteria include primal-dual tolerance thresholds. Shouldnt LA-ADMM use those instead of fixed t? With regards to linearized ADMM, Eqn 7, please comment on its relationship to Chambelle-Pock updates (https://hal.archives-ouvertes.fr/hal-00490826/document)

Reviewer 3



This paper exploits a local sharpness property to propose deterministic and stochastic (linearized) ADMM algorithms with adaptive penalty parameters that provably yield faster convergence rates than the respective algorithms with fixed parameters. The paper is well motivated, well written, and contains very interesting theoretical contributions. I recommend to accept the paper. The only complaint I have is that the experimental evaluation is very limited, particularly in the deterministic case. Instead of the nuclear norm minimization, I would have preferred the minimization of a polyhedral energy (ell^1 regularization + ell^1 data term) with a log-plot that verifies the linear convergence predicted by the theory. From the current experiment it is difficult to see if the adaptive step sizes yield a practical advantage. The authors could furthermore compare to simple adaptive schemes such as residual balancing. Finally, I am wondering about the computational overhead of the adaptive scheme over the classical ADMM: How would the plots in Fig. 1 look if the x-axis showed 'time' instead of 'iterations'? Minor comments: - The reference "Adaptive Relaxed ADMM: Convergence Theory and Practical Implementation" by Z. Xu, M. Figueiredo, X. Yuan, C. Studer, and T. Goldstein, CVPR 2017, could be of interest to you (but could be similar to [28] - I haven't read the paper in detail yet). - Please proofread the manuscript - there are several typos, e.g. l. 4 "are" -> "is", l. 13 rewrite "... scheme of lies at it is ...", l. 64 "stocahstic", Assumption 1(c) -> for all y?, l. 258 -> "the local sharpness parameter is \theta = 1", "enjoys" -> "enjoy".